# Examining public support for Ukraine's defense against autocratic aggression

**Lukas Rudolph** [1] ✉, **Fabian Haggerty** [2] **& Paul W. Thurner** [2]

Russia's invasion of Ukraine challenges the liberal international order and tests the capacity of Western democracies to maintain long-term military and financial aid for Ukraine in a foreign war. Understanding whether governments' pledges of resolve are backed by their citizens is crucial for the credibility of these commitments. Here we show, based on survey experiments with 10,011 respondents in the United States, the United Kingdom, Germany, France, and Italy, that these countries' publics share a similar pattern of preferences. In all countries, citizens strongly endorse Ukraine's sovereignty and self-determination while weighing human suffering and conflict escalation risk, but less so economic costs. However, within countries, attitudes are polarized: roughly one quarter of citizens with pro-Western orientations show firm resolve, whereas another quarter with anti-Western views remain largely indifferent to political outcomes for Ukraine. These divisions indicate that democratic party competition could constrain the unity and durability of Western resolve against autocratic aggression.

Liberal democracy is under pressure worldwide, with the number of democratizing countries at its lowest since the 1970s. Almost half of the global population lives in autocratizing contexts[1]. The dominant liberal world order, understood as a system of international institutions that promotes interconnectedness between states bound by shared rules[2], is contested[3]. This contestation includes the use of military coercion by autocratic states[4]. The struggle of Ukraine for its sovereignty and territorial integrity against Russian aggression since February 2022 exemplifies this challenge[5]. Western democracies (i.e., core members of the North Atlantic Treaty Organization (NATO) and their allies) are the central proponents of the liberal order and form the backbone of the coalition supporting Ukraine. However, their level of assistance to Ukraine has varied over time and between countries[6]. This indicates tensions in both their commitment to supporting Ukraine and their support of a more general rule-based international system[7,8].

We argue that Russia's war against Ukraine brings Western unity or disunity in terms of resolve against autocratic challenges to the fore[9]. However, how united Western countries are in their support for Ukraine is nontrivial, given the complex normative, economic, and strategic trade-offs in military and financial assistance to Ukraine.

Against this backdrop, we investigate mass public preferences regarding Ukraine support strategies in the US, UK, Germany, France, and Italy as the top five Ukraine support countries (with respect to total bi- and multilateral aid flows in 2022-23[6]). How strongly do citizens in Western democracies back continued support for Ukraine's defense? We propose that scrutinizing public opinion adds context to understanding the room to maneuver of governments in the Ukraine support coalition, helping to determine whether rifts in states' strategic approaches to Ukraine support occur despite, or because of, major rifts in preferences of their mass publics.

Contextually, the Western approach of extended deterrence to prevent Russia from invading Eastern European countries failed with the occupation of Crimea in 2014[10]. The full-out Russian attack on Ukraine in 2022 finally initiated a new step in this deterrence game. Western countries must credibly signal continuous financial and military assistance to a country not in a formal joint alliance[11] to restore Ukraine's sovereignty and deter further geostrategic ambitions of Russian and other autocratic[12] governments. For example, Aksoy et al.[13] found that the Russian invasion increased public support in China for military action against Taiwan, but not if Chinese citizens were informed of Western military aid to Ukraine.

[1]Department of Politics and Public Administration, University of Konstanz, Konstanz, Germany. [2]Geschwister-Scholl-Institute of Political Science, LMU Munich, Munich, Germany. ✉e-mail: lukas.rudolph@uni-konstanz.de

Therefore, we propose a new micro-foundation of autocratic containment and resolve of democratic citizens in this article. Theories of deterrence, including extended deterrence, usually adopt a unitary actor perspective. This perspective may be appropriate for autocratic contexts, but masks a potential vulnerability of democracies in deterrence games: the extent of domestic preferences that undermine resolve. A growing literature discusses the empirical role of resolve, i.e., of standing firm in crisis and war[14,15]. Resolve means threats to retaliate or provide military assistance to a protégé or ally (extended deterrence[16]) are considered credible by the adversary. The game-theoretic literature has prominently argued that democracies can exhibit greater resolve than autocratic regimes due to the audience costs of public commitments[14]. Recent work directly deduces this strategy to be applied by NATO countries: "By delivering the message that they are committed to supporting Ukraine, they aim to convince the Russians that allocating resources to fight in Ukraine is a worthless endeavor" (p. 1[17]). However, this strategy has been questioned both theoretically[18] and empirically[19] at other times[20]. Most importantly, the citizenry to which democratic leaders are accountable is usually quite heterogeneous in its preferences regarding their governments' crisis behavior. This heterogeneity can then be mobilized by parties taking different stances[21]. In addition, the threats or promises made by governments and parties publicly are usually ambivalent, leaving them discretion to reposition in response to changing conditions. Finally, these positions are partly endogenous to the anticipation of short-term electoral reactions. Subsequently, parties take non-uniform positions on the Ukraine war, where ideology and populist rhetoric matter for party positions across 29 countries[22], raising the question of how strongly this also reflects in citizens' preferences.

Hence, in this article, we focus on the extent and heterogeneity of preferences over Ukraine strategies among electorates of the US, UK, Germany, France, and Italy. Both weak overall support for defending Ukrainian sovereignty and territorial integrity and extensive domestic or cross-country heterogeneity would undermine the backing of democratic defenders in this situation of extended deterrence. In turn, strong and homogeneous support from these countries' electorates would credibly pressure government leaders to continue providing assistance. The degree of domestic dispute over the questions of crisis and within-conflict escalation, or protégé support, should therefore be an informative and objective indicator of the constraints to resolve that democratic leaders face. This perspective is in line with recent arguments that the credibility of deterrence signals by democracies depends on citizen preferences[23,24].

To test our argument, we conducted a tailored, extensively pre-registered (see Supplementary Discussion section 1.2) cross-country survey-experimental research design ($N = 10{,}011$) in the five major aid supplying countries supporting Ukraine (US, Germany, UK, France, Italy). We thereby shed light on this question with high internal validity while generalizing across populations from the divergent contexts of top NATO military powers; this is also theoretically an important addition to a literature which usually engages in single- or few-country studies with a heavy focus on the US case.

We make two contributions with our study: First, we integrate a literature investigating the decision-making of governments and citizens about foreign policy, noting that preference formation patterns probably diverge, as both attach different weights to the complex normative, economic, and strategic aspects of foreign policy[25,26]. We particularly relate to the literature on popular support for war, which shows that economic costs, moral constraints, and strategic risks are relevant factors in supporting military operations of a country[27,28], and that these factors can vary by context[29]. We extend this framework to decisions on external aid by third countries that have not been directly involved in a war fought on foreign soil. We propose that this can alter citizens' decision-making calculus, especially regarding the relative weight of domestic costs and foreign human suffering. Notably, our

framework allows us to shed light on citizens' preference formation regarding foreign policy more generally and to identify specific conditions under which public opinion can undermine or strengthen resolve to contain autocracy. Additionally, we propose that domestically, divisions in preferences could derive from the extent to which Western citizens themselves are rooted in the liberal, rule-based order promoted by Western democracies. This is a theoretical moderator that, to our knowledge, the foreign policy literature has not yet investigated. The case of Ukraine is particularly prominent in both regards: economic costs and strategic risks of escalation are relevant domestic concerns, while the human costs of this war and the immediate consequences of any agreement are only born by the population of the foreign country; at the same time, the Ukrainian fight against the Russian aggression is also a fight for the liberal, rule-based order. Second, given the reliance of the coalition governments that support Ukraine on electoral support, it is important to understand how the general public assesses such a scenario. The support Ukraine receives from Western countries provides a bound to Ukraine's domestic resolve, as Ukraine's military capabilities otherwise cannot keep up with those of the aggressor in the (as of 2025) ongoing war of attrition[30]. But at the same time, military aid can only be effective as long as the Ukrainian population also supports continued fighting. Hence, preferences within Ukraine also provide a bound to the resolve that Western countries or their populations can exhibit. Our results directly address this dual dependence, as our research design is directly comparable to related work by Dill et al.[31,32]. They find that the attacked Ukrainian population – in line with its government—is by a substantial majority willing to fight "at any cost" against the Russian aggression, accepting both strategic risks (i.e., nuclear escalation) and human costs (among military and civilians) for upholding their full sovereignty and, albeit weakening over time, their territorial integrity. Descriptive evidence from surveys with European and US citizens indicates that Western populations could exhibit much more nuanced preferences[33,34]. However, as we know from the survey methodological literature[35], only well-designed survey experiments allow the proper extraction of the preference structure of survey participants faced with multi-dimensional trade-offs, as in our case. Furthermore, an experimental approach eases the pervasive issue of social desirability in single-item surveys[36], particularly when survey participants assess valence issues, as in our case. We therefore provide a survey-experimental exploration of mass public attitudes among the Ukrainian support coalition for their resilience and resolve against autocratic aggression, thereby informing both scholarly debate and public policies. This comparison does, at the same time, allow for a contrast of preference formation between directly war-affected and generally allied but not directly affected populations, an issue we return to in the discussion.

Our results show that the publics of the major countries in the Ukraine support coalition support Ukrainian territorial integrity and sovereignty on average, but take human costs and strategic risks, but not economic costs, relevantly into account. Our cross-country comparisons indicate similar levels of resolve across the five countries, slightly stronger among US, UK, and German citizens. However, sub-group analyses reveal substantial heterogeneity in resolve within countries, depending on individuals' attitudes towards transatlanticism (as an indicator of support for the liberal world order). A quarter of the citizens exhibit very firm resolve, similar to the Ukrainian population[31], while another quarter is largely indifferent to the political outcomes of the war for Ukraine. Second, in a subsequent vignette experiment, we investigate the underlying reasons for these preference patterns. We find that offensive military aid packages are associated with increased perceptions of Russian containment, but also potential escalation. Importantly, respondents with pro- vs. anti-Western attitudes show opposing perceptions of the consequences of economic and military aid, consistent with divergent favorability of Ukraine strategies with strong resolve.

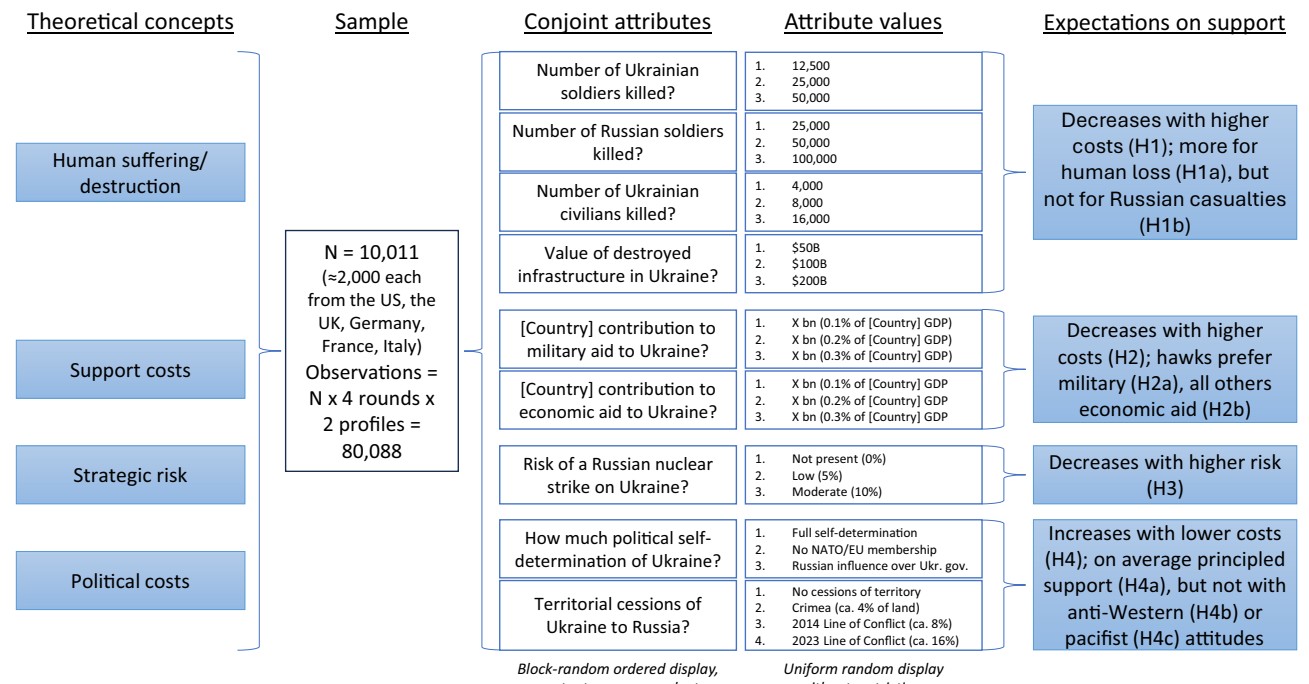

**Fig. 1 | Our argument for the conjoint experiment proposes that four core aspects of Ukraine support strategies shape mass public preferences in Western democracies: political costs, i.e., the extent of political containment of the Russian aggression; the human cost, i.e., human suffering and destruction through the war; support costs, i.e., domestic financial costs of aid; and strategic risks of escalation associated with support—where the latter three are cross-pressures for strong support of pro-Ukrainian war outcomes and their** weight in respondents' calculus indicates citizens' resolve to contain the Russian aggression. We investigate preference patterns by a binary profile choice experiment with 10,011 respondents from the five major aid-supplying countries supporting Ukraine. Nine attributes with three to four levels each operationalize our theoretical concepts, for which we derived concrete pre-registered expectations.

## Results

We were careful in designing a study that has both strong external validity—given that we study five populations with a complex quota design proven to match the current structure of political preferences in these five countries—and strong internal validity—by drawing on conjoint and vignette survey experiments that allow the identification of causal effects. We first present the setup and underlying motivation for the conjoint experiment, and the corresponding results. Then, the vignette experiment follows.

### Four dimensions of mass public preferences on strategies for Ukraine

In our conjoint experiment, we confronted respondents with a forced-choice paired-profile conjoint design for political strategies with respect to Ukraine support[35]. We use nine attributes that correspond to the four theoretically derived dimensions of our experiment: resolve (i.e., political costs), domestic financial burden (i.e., support costs), human costs, and strategic risk. Note that we partially replicate the design of Dill et al.[31]. Each attribute has three to four uniformly randomized levels, carefully selected to represent the range of real-world outcomes as discussed in the media environment at the time of our survey (see Methods section "Set-up of the conjoint experiment" and the pre-registration). Figure 1 presents an overview of the core theoretical concepts together with the design of our conjoint experiment and a summary of our pre-registered expectations. Supplementary Discussion section 1.2 presents details on how this summary relates to our pre-registration and notes deviations from it.

We derive these expectations from a broader argument about how citizens solve the underlying trade-offs entailed in Ukraine support strategies. We first note that we have good reason to expect, on average, principled support by Western publics for a containment of the Russian aggression (lower political costs), i.e., Ukrainian

sovereignty and territorial integrity. Fending off the Russian aggression would not only signal resolve, it would, by definition, be defending the underlying fundamental norms enshrined in the liberal world order and the ensuing international law: the principle of rightful self-defense against unlawful aggression and remedy against the alleged Russian violations of international humanitarian law accompanying this war[37]. Defense for such fundamental normative principles is in line with basic moral codes, also enshrined in domestic legal codes, which should hence see relevant support in the citizenry—as is the case regarding citizen's support for war or humanitarian intervention[26,38], where approval for troop deployment increases substantively with normatively justified objectives[39,40]. This is also what descriptive polling of Western public opinion on Ukraine support indicates (see Supplementary Discussion section 1.1).

But how far does this support go? We are particularly interested in three thematic cross-pressures of supporting Ukrainian war aims, which could undermine such resolve: human costs, economic costs of support, and strategic risks associated with Ukraine support strategies. Their relevance could display heterogeneously both within populations and between states, and provide insights on underlying motivations of citizens.

Human costs could undermine resolve. From the point of view of moral theory, Western populations do not bear these themselves and therefore do not have moral legitimacy to disregard them[31]. Moreover, the principle of proportionality (see Dill et al.[31] for a summary), whereby even rightful self-defense needs to be proportionate with respect to the human costs of war[41,42], implies that the populations of Western countries should decrease support for Ukraine strategies if related to an increasing death toll among the war parties. Related literature shows that citizens strongly decrease their support for war when faced with the death and destruction caused by military engagement, both among their own but also among foreign

populations[29,43]. Likely from this perspective, we also observe a principled rejection of arms transfers by pacifist factions of Western societies[25]. At the same time, human costs seem to be weighted differently depending on which side of a war they accrue on[44], and death among foreign (civilian) populations has been, for the US example, found to matter relatively little, with "the public's commitments to proportionality [...] heavily biased in favor of protecting American soldiers and promoting US national security interests" [ref. 43, p. 548]. It is therefore an open but important empirical question how strongly citizens consider death among the militaries of the war parties and the Ukrainian citizenry when forming preferences. Likely, citizens consider death tolls, but less so among the Russian side.

The monetary costs of military and economic aid to Ukraine could also undermine resolve. Foreign aid spending implies domestic opportunity costs, i.e., assistance affects the sustainability of domestic budgets and competes with domestic spending on social welfare or other security measures. Related research shows that such opportunity costs can decrease support for public spending[45,46]. More broadly, spending aversion also reflects in isolationist tendencies documented for a relevant fraction of citizens in both the United States and Europe[47–49]. At the same time, mutually delivered benefits of aid can offset cost arguments[50]. In addition, citizens seem to assess certain types of aid differently, in particular economic aid compared to military aid[51], which likely depends on their principled foreign policy stances[52]. To what extent the financial costs of Ukraine aid are taken into account by citizens is, therefore, again an empirical question. Likely, higher costs decrease support, with underlying foreign policy values shaping preferences over economic vs. military aid.

Finally, security risks, i.e., the threat of military escalation, could undermine the resolve. In our case, citizens could be particularly concerned about the risk of nuclear escalation, which can constitute a strong barrier to engagement[53]. Just as at the end of the Cold War, when Western peace movements in the aftermath of the NATO double-track decision in 1979 constrained NATO governments' deterrence of the Soviet Union[54], descriptive evidence indicates a strong support for appeasement politics vis-à-vis the Russian war in Ukraine for some European societies[34]. More generally, the foreign policy literature notes a public aversion to supporting intervention with potential to escalate to the loss of domestic lives[55]. Therefore, we expect that the perceived risk of nuclear escalation can serve as a barrier to Ukrainian support.

Scrutinizing preference formation along these three dimensions is worthwhile in its own right – comparing preference patterns with the war-support literature and with those among the Ukrainian population. However, we also conceive of citizens' support for Ukrainian territorial integrity and political sovereignty, despite these trade-offs, as an indirect measure of the resolve of Western mass publics against autocratic challenges. Later sections expand the above argument to within-country heterogeneity, to concrete types of military and economic aid, and to perceived consequences of such aid for war intensity, Ukraine, and the domestic context.

### Western publics support Ukrainian territorial integrity and sovereignty, but take human costs and strategic risks into account

Subsequently, we present our empirical estimates. Our main quantity of interest are Average Marginal Component Effects (AMCEs)[35] as well as Marginal Means (MMs)[56] for subgroup analyses (for details, see Methods section "Estimation strategy and robustness", also discussing estimates from Average Component Effects (ACPs, Ganter[57]) and Average Feature Choice Probabilities (AFCPs, Abramson et al.[58]) given recent debates on AMCE interpretation—these show the same pattern of results as our AMCE/MM plots presented below).

Figure 2 reports AMCEs for the full sample. Attribute levels are ordered such that the theoretically most favorable levels serve as the baseline. First, as can be seen, across our four dimensions (separated by dashed lines), higher levels of attributes consistently and significantly ($p < 0.001$, except for Russian casualties, with $p = 0.139$ also for highest level 100,000) lead to lower choice probabilities on average. This is in line with our expectation of a decrease in support for strategies with larger human (H1), economic (H2), or strategic costs (H3), or with a lower containment of the Russian aggression (larger political costs) (H4). Next, we interpret the relative size of the AMCE spread within attributes—the difference between the baseline and the highest level—as the relative weight of each attribute in respondents' decision-making. Given that we have three levels in all but one attribute, we can directly compare their relative size[57]. The strongest determinants of support are the containment of aggression, human costs, and strategic risks.

This implies two key findings: on the one hand, we observe that average support for Russian containment is by no means unconditional (against our expectation in H4a). Respondents place emphasis particularly on the human costs of war (with high extent of Ukrainian civilian and military deaths substantively decreasing choice probabilities, by 8 percentage points for the latter, and 10 percentage points for the former comparing maximum expression to baseline) and nuclear escalation potential (with a 10% nuclear strike risk decreasing choice probabilities by 9 percentage points, compared to baseline) at comparable levels to extensive containment (with coefficient estimates for a high extent of concessions and loss of self-determination, with both -11%-points compared to baseline, being only marginally larger). While attributes are on different scales, we presented the minimum-to-maximum range of realistic outcomes for potential strategies (see Methods section "Attribute and attribute-level selection"). This indicates, overall, a conditional calculus that limits the extent of resolve with higher human costs and strategic risks, though not economic costs.

On the other hand, the costs of aid feature low in citizens' preferences formation. To explore this further, we compare the effect of cumulative financial costs of aid, ranging from 0.2% to 0.6% of GDP in our experiment, to the combined expression of containment attributes, on choice probabilities. The results are reported in Supplementary Fig. 4, model 4: monetary costs of 0.6% of GDP of combined military and economic aid (corresponding to, e.g., $120B for the US) see a decrease in choice probability of 5.1 percentage points, while combinations of maximum concessions and Russian influence see a decrease of 22 percentage points. This implies a substantively higher— 22/5.1 = 4.3 times—weight in respondents' calculus. Note that we find no statistically significant interaction effects, i.e., we find no credible evidence that attribute levels of both dimensions affect choice in a non-linear way (Supplementary Fig. 4, models 2 (categorical interaction of cost attribute) and 3 (metric interaction of cost attribute, to increase power)). Hence, assuming the appropriateness of linear extrapolation, only if aid would reach 4.3*0.6% = 2.6% of GDP ($703B for the US), would choice probabilities decrease as they do with the change from maximum to minimum containment of Russia from our experiment.

Next, we interpret the pattern of preferences among attributes of the same dimensions: Concerning the economic dimension, we do not detect statistically significant or substantive differences in the changes to the baseline between the high degree of military and economic aid on average ($p = 0.366$, testing both coefficients against each other). Against our expectation in H2b, we therefore find no credible evidence that economic (compared to military) aid sees stronger average support. Concerning human suffering and destruction, on the one hand, we find a much weaker reaction to the loss of Ukrainian infrastructure compared to human death ($p < 0.001$, as expected with H1a). On the other hand, we identify substantive differences in choice behavior depending on the combatant status or nationality of the casualties. Based on linear extrapolation, preferences show no statistically

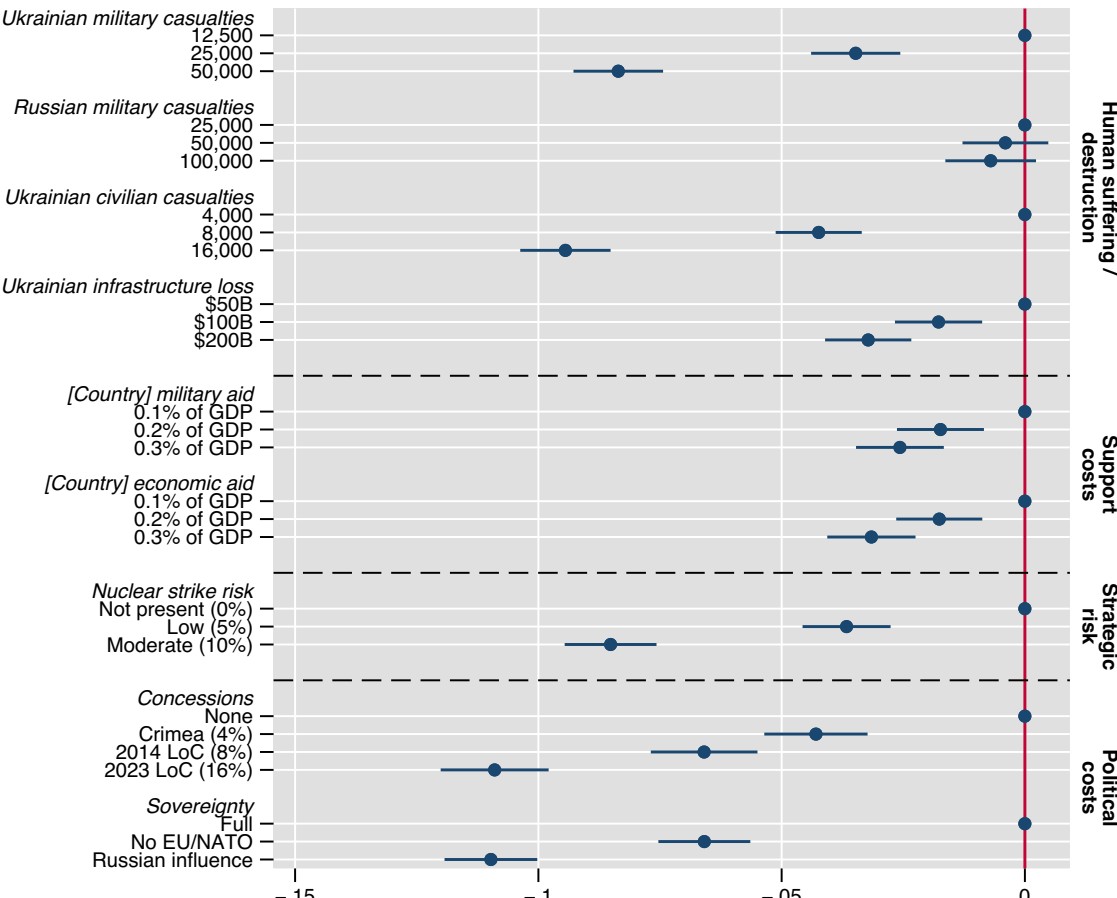

**Fig. 2 | Ukraine support strategies which do not achieve good political outcomes for Ukraine, but also with larger human suffering among the Ukrainian side, or risk for nuclear escalation, see lower choice probabilities on average.** The figure presents Average Marginal Component Effects (AMCEs) for the binary paired profile conjoint-experimental choice task. AMCEs represent the average causal effect on the probability of choosing a strategy (response scale: 0/1) when changing the level of an attribute within a profile from a baseline to another level while averaging over the randomized distribution of other attributes. Reading example for the bottom-most coefficient: the probability of choice is 11%-points lower for profiles depicting "Russian influence" compared to "Full sovereignty", averaging over the distribution of the remaining attributes. Error bars denote 95% confidence intervals from respondent-clustered standard errors. $N = 80,088$ observations from $N = 10,011$ respondents. Estimates presented are derived from linear regressions of choice on attribute-level indicators with the theoretically most favorable levels as baseline. Supplementary Table 7 presents corresponding statistics.

significant relation to the death of Russian soldiers—with an insignificant decrease ($p = 0.136$) of 0.0007 per 10,000—, but respond strongly to the death of Ukrainian soldiers—with a decrease in choice probability of 1.6 percentage points per 10,000, $p < 0.001$—and even more strongly to Ukrainian civilian deaths—with a decrease of 6 percentage points per 10,000, $p < 0.001$—(as expected with H1b). Concerning the containment of aggression, the minimum-to-maximum range of concessions and of sovereignty appears to be of similar importance to respondents, with an 11%-point spread each.

**Cross-country comparisons indicate comparable preference patterns**

Next, we provide systematic comparisons between the five countries in our study. This is important because it relates to the unity or disunity of Western publics, which has been extensively discussed as a prerequisite for effectively containing and deterring Russia[34,59]. Figure 3 displays MMs for the choice task by country subsamples. We note the following patterns, with differences we describe as statistically significant below being significant at the 5% level based on Bonferroni-adjusted confidence intervals (given the large number of comparisons, see Supplementary Table 10).

First, regarding Ukrainian infrastructure damage, and death of Ukrainian military personnel or civilians, we observe largely similar preference patterns between countries, besides a slightly higher penalty for high civilian loss in the UK sample (statistically different to the Italian and German sample). Concerning Russian military casualties, preference patterns diverge between the Italian, German, and French samples, decreasing with higher casualties, while for the US and UK samples, we observe higher choice probabilities with higher casualties, significantly so for the UK (compared to the French and Italian sample). However, note that the relevance of this attribute is substantively small in all samples. In sum, we find little credible evidence that the human costs of war are being taken into account differently across countries. This applies, second, also to the consideration of economic costs, where we find no statistical differences for the favorability of attribute levels under Bonferroni correction. Third, this also applies to strategic risks, where citizens of the five countries mostly respond similarly, except for a slightly lower favorability in the UK sample (coefficient of 0.44) for moderate risk (statistically different from the French sample with 0.47) and a higher level of acceptance in the UK sample for no risk (with 0.56, statistically different from the German sample with 0.52). Finally, concerning the political containment of aggression, all countries again show a similar structure of lower favorability for more concessions/less sovereignty for Ukraine. However, the Italian sample is least negatively reactive with respect to higher concessions and least positively reactive to no concessions; the German and UK samples

show the inverse pattern. Differences are of substantively relevant size: with choice probabilities 0.47 for high concessions for the Italian sample, significantly different from 0.43 for the German and UK samples; and 0.51 for low concessions, significantly different compared to 0.57 for the German and UK samples. In addition, regarding political sovereignty, the UK (coefficient of 0.44) and German (0.43) samples show the lowest choice probabilities for the toleration of Russian influence, differing significantly from an attenuated response in the French sample (0.47) and just insignificantly (0.46) in the Italian sample. Moreover, the UK sample (coefficient of 0.59) has the highest choice probability for full sovereignty, significantly different from the Italian (0.54), German (0.56), and French (0.53) samples; the US sample (0.57) has the second highest, significantly different from the French sample.

In sum, we find little credible evidence of a difference in the reactions of the country samples to human (on the Ukrainian side), economic, and strategic costs. In turn, the full sovereignty and territorial integrity of Ukraine show higher favorability in the UK, German, and US samples compared to the French and Italian samples. The spread is particularly pronounced for both attributes when comparing the UK and Italian samples. The question of why this is the case would require extensive linkage of our survey data with data on elite (e.g., party, executive, or legislative representatives) rhetoric and/or media coverage of the Ukraine war among these five countries at the time of our survey. This question is important, but it is beyond the scope of this article, so we leave it to future research. We still emphasize, however, that the overall pattern of MMs we identify with our experiment aligns across the five countries under study—implying a similar general structure of preferences for these countries' publics.

## Subgroup analyses reveal heterogeneity within countries

In the next step, we dive deeper into potential polarization within the country samples. Extending our argument described at the beginning of the Results section, scholarly work proposes that citizens' attitudes are determined by fundamental values. Regarding support for war, the literature prominently shows that foreign policy values[47] or attitudes towards war and peace[60] are central determinants of citizens' preferences and predict the relative importance of normative, economic, or strategic considerations. We therefore proposed assessing heterogeneity along these lines in our pre-registration. However, given we inquire (Western democracies') resolve against (Russian) autocratic aggression, we likewise pre-registered (comp. H4b) that the extent to which Western citizens themselves are rooted in the liberal, rule-based order promoted by Western democracies is likely at the core of within-country heterogeneity. This is in line with broader arguments that the contestation of the current liberal world order not only stems from actors outside the "West", but also from within, i.e., from factions in Western societies, particularly far-left and far-right movements[61]. Notably, the backbone of the current liberal world order is the transatlantic security framework based on a US-led NATO alliance seeking to expand and defend liberal internationalism. To capture whether respondents align with or oppose liberal internationalism, we therefore distinguish respondents by whether they support or renounce transatlanticism. These value dispositions are a much more general account of preferences regarding the role of the Western alliance compared to our concrete assessments of Ukraine strategy. We conducted extensive exploratory analyses comparing attitude heterogeneity along the lines of foreign policy values, attitudes towards war and peace, and heterogeneity by pro-/anti-Western attitudes. These are based on composite indices derived from a principal components analysis (PCA) on multi-item batteries tailored to measuring these values/attitudes. We then split all respondents into quartiles based on the PCA-derived dimensions and estimate MMs to assess the extent of heterogeneity across the three types of value predispositions (see Methods section "Non-experimental variables and measures" for

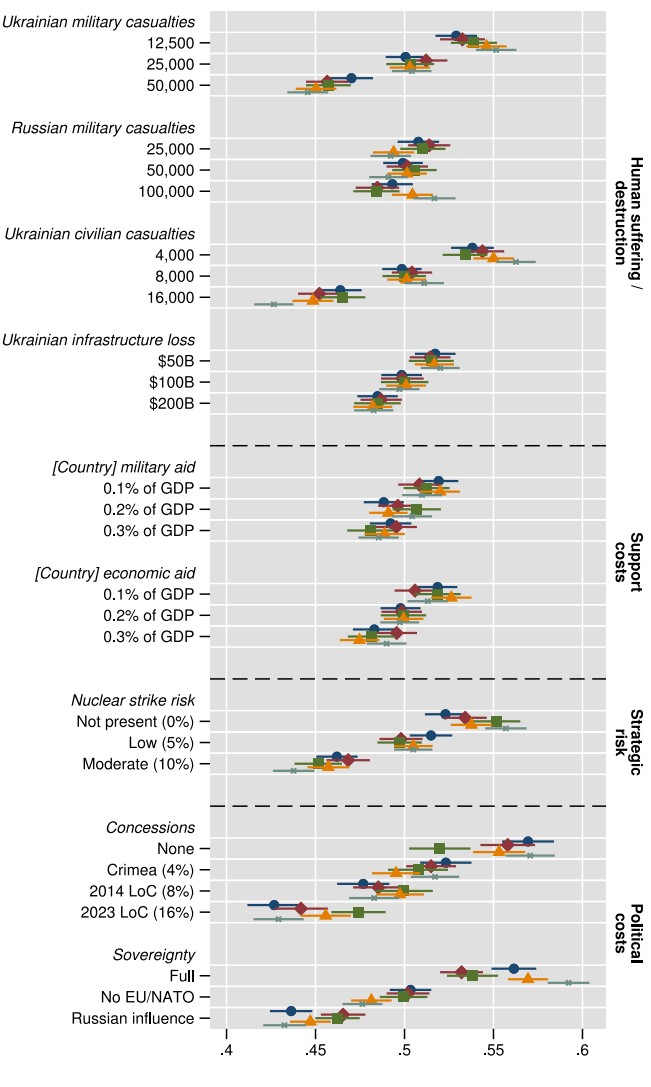

Fig. 3 | Across the five countries, respondents show a similar preference structure on average, with country-heterogeneity particularly regarding political outcomes for Ukraine. Marginal means (MMs) for the binary paired profile conjoint-experimental choice task by country. Averaging equivalently as AMCEs, MMs are the probability of choosing a choice option with randomization of an attribute to a specific level. Within subgroup and attribute, differences between MMs correspond to AMCEs. MMs (scale 0/1) by definition average 0.5, with values above (below) 0.5 indicating attribute levels increasing (decreasing) profile favorability. Reading example for bottom-most UK coefficient: probability of choosing profiles depicting "Russian influence" is at 43%, i.e., disliked, averaging over the distribution of the remaining attributes. Error bars denote 95% confidence intervals from respondent-clustered standard errors. Overall $N = 80{,}088$ from $N = 10{,}011$ respondents, with 16,008 to 16,024 observations by country sample (blue circles: German (DE); red diamond: French (FR); green square: Italian (IT); yellow triangle: United States (US); gray cross: United Kingdom (UK)). Estimates presented are predictions for choice from linear regressions of choice on attribute-level indicators within country subgroup. Supplementary Table 8 presents corresponding statistics, Supplementary Tables 9 and 10 with Bonferroni-adjusted confidence intervals.

details). We find that pro-/anti-Western alignment is the major divide between citizens' positions regarding Ukraine policies. Hence, we focus on this heterogeneity below and return to heterogeneity by foreign policy values and attitudes towards war and peace at the end of this section.

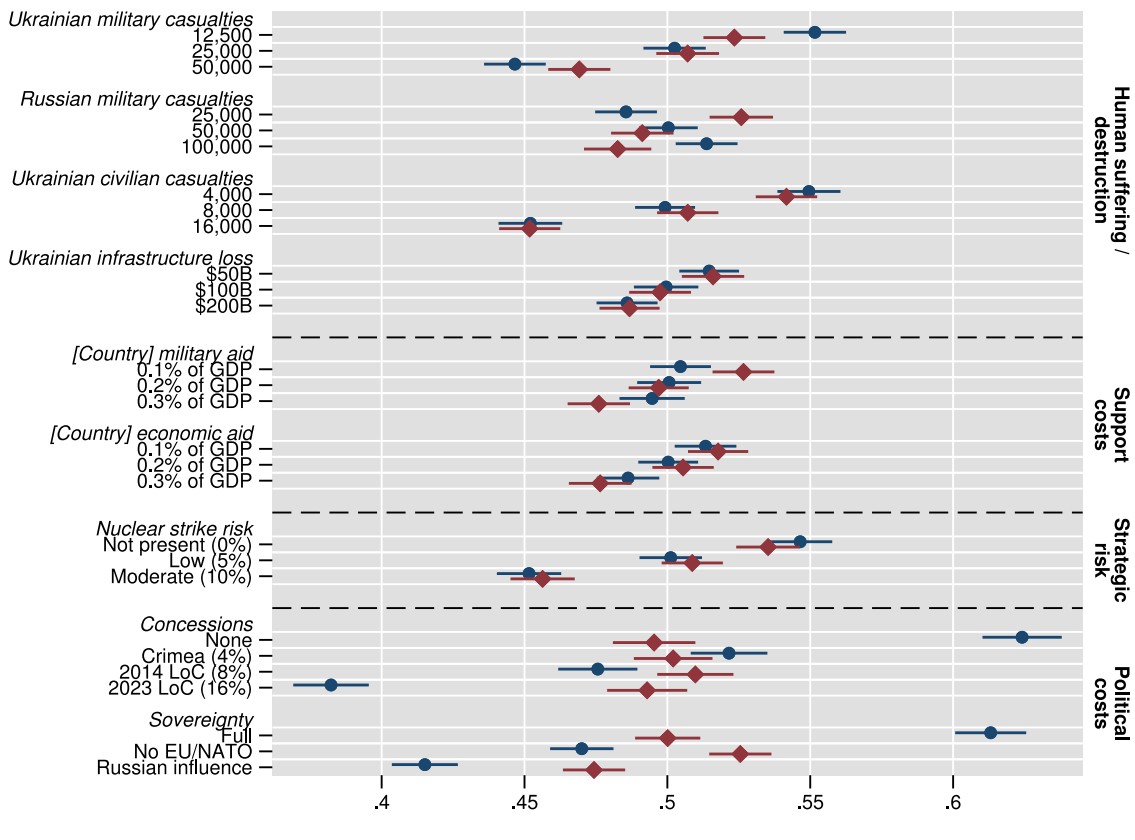

Strong pro−Western attitudes (Quartile 1) ◆ Strong anti−Western attitudes (Quartile 4)

**Fig. 4 | Respondents with strong pro-Western attitudes show much higher favorability for Ukraine strategies with political outcomes beneficial for Ukraine, to which respondents with strong anti-Western attitudes are largely indifferent, while both groups show no statistically significant differences regarding most other attributes.** Marginal means for the choice task by subgroups of the first and fourth quartiles of pro-/anti-Western attitudes. Results for all quartiles are presented in Supplementary Fig. 5. Error bars denote 95% confidence intervals from respondent-clustered standard errors. $N = 37,408$ overall observations from $N = 9352$ respondents, with $N = 18,936$ observations for the subgroup with strong anti- (red diamonds) and $N = 18,472$ observations with strong pro-Western attitudes (blue circles). The estimates presented are predictions for choice from linear regressions of choice on attribute-level indicators within attitude subgroups. Supplementary Table 11 presents corresponding statistics, Supplementary Tables 12 and 13 with Bonferroni-adjusted confidence intervals.

Figure 4 presents estimates for respondents in the first quartile (strongest pro-Western, blue circles) and the fourth quartile (strongest anti-Western, red diamonds). Supplementary Fig. 5 presents results for intermediate groups whose preferences are located between these extremes. Given the large number of comparisons, we interpret differences between groups as statistically significant if $p < 0.05$ based on Bonferroni-adjusted confidence intervals (see Supplementary Table 13). Our findings are threefold. First, we find no credible evidence that choice probabilities diverge for several dimensions, in particular for casualties among Ukrainian civilians, infrastructure damage in Ukraine, economic and military aid, and nuclear risk. We still point out that, though group differences are not statistically different under Bonferroni correction, for respondents with anti-Western attitudes, larger amounts of both military and economic aid statistically significantly reduce choice probabilities, while for respondents with pro-Western attitudes, we observe a statistically significant reduction only for the latter. Second, a significant divide in choice probabilities appears for the remaining attributes: regarding military casualties, respondents with pro-Western attitudes more strongly deselect choice options with high Ukrainian (coefficient 0.45), but more likely select choice options with high Russian casualties on average (0.51); respondents with anti-Western attitudes react with decreased acceptance to either, although attenuated regarding Ukrainian casualties (0.47). Third, and most importantly, a substantively strong divergence of preferences is induced by the attributes of territorial concessions and sovereignty. Regarding the former, with anti-Western attitudes,

respondents do not respond in a differentiated way to varying levels of territorial concessions (all around 0.5, hence, respondents are indifferent between levels). Contrary, with pro-Western attitudes, we see a strong spread of preferences, with a 25%-point difference in no concessions and the current frontline. Likewise, any sacrifices or blurrings of self-determination are strongly penalized among respondents with strong pro-western attitudes—here, we observe a 20%-point difference between full sovereignty and a hypothetical Russian influence for this group. In turn, respondents with anti-Western attitudes demonstrate a slightly higher choice probability for a neutral status of Ukraine (coefficient 0.53), while, on average, slightly disfavoring actual Russian influence over the Ukrainian government (0.47). Hence, for the quarter of respondents with firm pro-Western attitudes, containment of Russia seemingly dominates the choice table, while those with anti-Western attitudes want fighting to end irrespective of territorial gains of Russia and with a neutral status of Ukraine. Taken together, this indicates that, at least for this relevantly sized subset of respondents with firm pro-Western attitudes, our expectation H4a holds: this subgroup shows principled support for upholding the sovereignty of Ukraine. As the converse of this finding, the presented evidence is consistent with expectation H4b that respondents with strong anti-Western attitudes barely care about political outcomes for Ukraine.

Subsequently, we explore in non-preregistered analyses who these respondents with strong pro-/anti-Western attitudes are. First, we consider descriptive support for the statements underlying the PCA score in both groups. Almost all of the respondents with strong

pro-Western attitudes reject that they can understand Russia feels threatened (89%), reject that US imperialism is a danger to world peace (83%), and report a positive stance towards NATO (87%); while a vast majority of the respondents with strong anti-Western attitudes agree that Russia feels threatened (74%), agree on US imperialism (76%) and report a negative (50%) or torn (38%) stance towards NATO. Second, Supplementary Fig. 6 indicates that both the composite PCA index and its three components show a U-curved relationship with the standard left-right axis: scores are higher for both the extreme right and the extreme left, pointing to a specific coalition potential of anti-democratic publics within the Western democracies, and to threats to resolve from both the radical left and right. Third, Supplementary Table 4 goes into more detail on who these anti-Western respondents are. Here, we regress anti-Western preferences (PCA index) on respondents' sociodemographics, as well as political attitudes and values. According to model 1, we find that anti-Western attitudes are relevantly associated with country of origin (lower in the UK, higher in Italy, compared to France, Germany and the US ($p < 0.001$)), with age (lower in age cohorts above 45, $p = 0.029$, and above 60, $p < 0.001$, compared to under 31), with education (lower for high- compared to low-educated respondents, $p < 0.001$), higher with worry over the personal ($p = 0.001$) or country-wide state of the economy ($p < 0.001$, compared to no worry), associated with religious affiliation (lower with Protestant ($p < 0.001$), higher with Orthodox ($p = 0.003$) or Muslim faith ($p < 0.001$), compared to no stated religious affiliation or Catholic faith); there is no statistical evidence of an association with gender or work status (ceteris paribus). Adding political attitudes to the regression (model 2), the above-mentioned associations remain robust. Ideology again shows a U-shaped relationship (though significant ($p < 0.001$) only for the political right), while the stated turnout for the last general election relates negatively to anti-Western attitudes ($p < 0.001$); political interest shows no statistically significant relationship. Last, adding additional value configurations of respondents to the regression, foreign policy values are, ceteris paribus, associated relevantly: isolationist attitudes correlate positively ($p < 0.001$), while militant interventionist ($p < 0.001$), but also cooperative internationalist ($p = 0.031$) values correlate negatively with anti-Western attitudes (model 3). Attitudes towards war and peace (with higher values indicating pro-war or pro-peace attitudes) correlate positively with anti-Western attitudes ($p < 0.001$) (model 4). Compared to the standard deviation of the dependent variable, effect sizes are in-between 8% (age group 46–60, model 1) and 49% (Orthodox faith, model 2) of this standard deviation. Hence, they can reach a relevant magnitude. Taken together, these results clearly indicate, on the one hand, that idiosyncratic country-level factors beyond the set of correlates we used lead to variation in resolve against aggression between the five countries; on the other hand, within countries, the shown associations provide some context for which societal coalitions for pro- and anti-Ukraine support could form.

Finally, we briefly note that attitudes towards war and peace, as well as foreign policy values, also matter for observed preferences. On the one hand, when differentiating respondents into respondents with pacifist and hawkish attitudes (see Methods section "Non-experimental variables and measures"), we find that the former show a relevantly higher consideration of human casualties, diminishing their overall resolve (in line with expectation H4c; see Supplementary Fig. 11, left panel). In turn, hawkish respondents show no statistically significant reactions to changes in military aid (in line with expectation H2a, see Supplementary Fig. 11, right panel). More broadly, foreign policy values and attitudes towards war and peace also structure how citizens view foreign policy, but not to the extent we see with pro-/anti-Western attitudes (see Supplementary Figs. 11 and 12). Supplementary Discussion section 1.2 compares these analyses in more detail to our pre-registration.

## Preferences on specific types of military and economic aid and their perceived consequences

In a last step, we extend our results on abstract Ukraine strategies with original evidence regarding (dis)agreement of respondents to providing specific types of aid, as well as perceived consequences of such aid. This not only directly informs policy debates in the countries under study on how to assist Ukraine concretely, but gives us an indication by which means and for what reasons respondents want to stand firm with Ukraine or not. We inquired into the two core dimensions of aid, military and economic, and presented respondents with the most prominently discussed aid options in public debate at the time of our survey.

We chose a 1 ∗ 6 vignette design for this question. Our aim is to causally compare stated support levels and stated perceived consequences of aid by the respondents' government to Ukraine. Given that respondents are asked about one vignette only, our design prevents spillovers in respondents' assessments of one type of aid to another. This experiment was placed directly after the conjoint experiment. Figure 5 provides details on our design and pre-registered expectations.

Our theoretical justification for this vignette lies in the heavy contestation of this aid within Western democracies. Western governments provide this aid with the intention of enabling Ukraine to defend itself, i.e., to increase its winning chances and end the war. We propose that citizens will likely agree to aid if they care about the outcomes in Ukraine. Hence, we also pre-registered as baseline expectation general support for offensive aid on an absolute level, though decreasing with aid's offensive potential, expecting that citizens perceive offensive aid to increase Ukraine's chances on the battleground, but also lead to higher human and economic costs and strategic risks.

Note that given the results of the conjoint experiment, respondents with anti-Western attitudes do not seem to care about outcomes for Ukraine. Hence, they should also disfavor military aid (while respondents with pro-Western attitudes should agree—and all the more so, the more they perceive its consequences as beneficial for Ukraine).

However, the conjoint experiment also indicates that respondents take human costs and strategic risks relevantly into account. It is an empirical question, therefore, to what extent respondents simultaneously perceive negative consequences regarding escalation risk and human suffering (and also domestic economic opportunity costs), and whether overall negative consequences on these dimensions can motivate disagreement. If negative consequences have a large weight, this would clearly undermine Western governments' ability to supply robust aid to Ukraine and thereby directly diminish their potential resolve, restricting them to certain aid types.

Note that we cannot causally identify why respondents agree with certain types of aid. Perceived consequences are likely causally prior to agreement to aid, but could also, e.g., via motivated reasoning, be endogenous to support.

## Offensive military aid relates to increased perceptions of Russian containment, but also escalation potential

Figure 6 presents evidence on our first dependent variable: agreement with providing certain types of military or economic aid. As can be seen, respondents are, on average, close to the midpoint of the agreement scale (1: disagree fully; 7: agree fully) for the three military aid items, and thus also tentatively agree with military aid with offensive potential (in line with expectation H5). In particular, agreement increases statistically significantly with comprehensive military aid, i.e., with additional tanks and fighter jets ($p < 0.001$), compared to air defense only—contrary to our expectation in H6. Economic sanctions see the relatively strongest agreement (coefficient 4.7), while backing

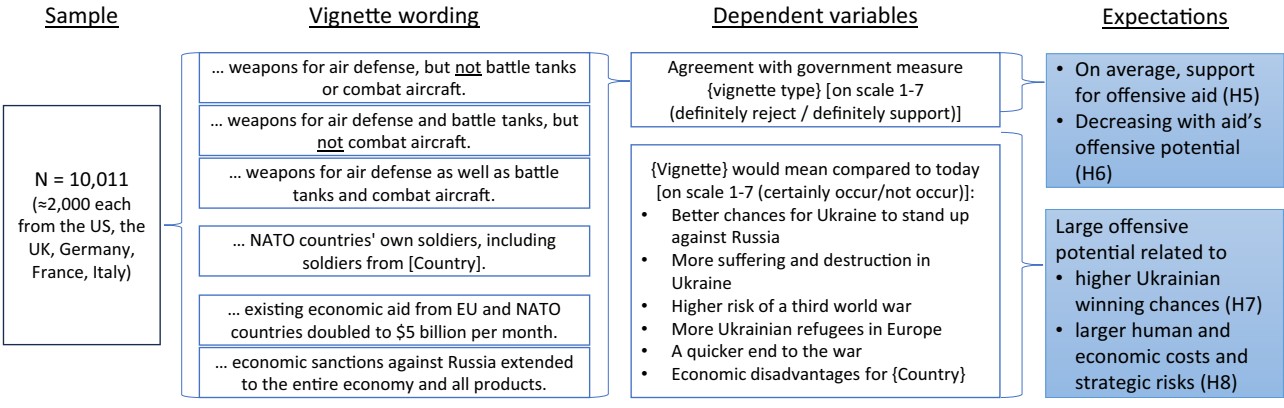

Uniform random draw of one vignette per respondent.          Random ordering of response items and scale order.

**Fig. 5 | Our argument for the vignette design links differences in agreement with aid types to perceived consequences of aid domestically and in Ukraine.** We investigate preference patterns by a 1*6 vignette presenting respondents with four types of military and two types of economic aid/sanctions for the 10,011 respondents from the five major aid-supplying countries supporting Ukraine. The four types of military aid (from air defense only to also battle tanks and combat

aircraft) allow for comparing military aid with increasing offensive potential. We examine two sets of dependent variables (DVs): first, (dis)agreement with the provision; and second, a battery of six potential consequences of providing this aid. We derive concrete pre-registered expectations of how aid types are linked to support and perceived consequences.

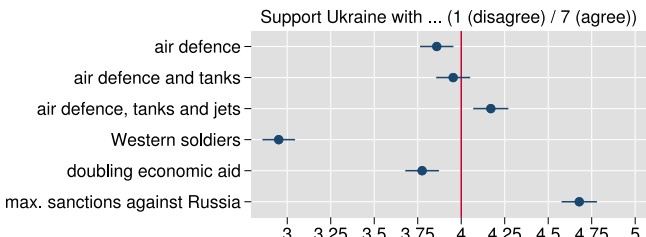

**Fig. 6 | Respondents agree, on average, with their governments' provision of offensive military aid including combat aircraft, and increasing sanctions against Russia, but are close to the scale midpoint or in disagreement for less offensive forms of military aid, doubling economic aid, and the provision of ground troops.** Figure displays marginal means for agreement to provide six different types of military and economic aid to Ukraine based on the split-sample vignette (scale 0 (no) to 7 (full agreement)). Error bars denote 95% confidence intervals from robust standard errors. $N$ = 8596 respondents. Estimates presented are predictions from linear regressions of aid agreement on vignette conditions. Supplementary Table 14 presents corresponding statistics.

for troop support (3.0) is the lowest. A doubling of economic aid (3.8) is slightly less approved compared to the three military aid packages, significantly so for air defense and tanks ($p$ = 0.011) and air defense, tanks, and combat aircraft ($p$ < 0.001). As indicated by Supplementary Fig. 8, this pattern varies by country: most prominently, fighter jets compared to air defense only see relatively stronger agreement in the UK and US; tanks are agreed to more in Germany and the UK (compared to air defense only).

Crucial to this section is how respondents perceive the consequences of specific military or economic aid. The question wording ensured that these were elicited relative to actual aid levels at the time of the survey. We first compare military aid to Ukraine. As indicated in Fig. 7, on average and compared to air defense alone, the delivery of tanks and fighter jets goes hand in hand with the respondents perceiving such deliveries as increasing the chances of winning for Ukraine (up 0.47 scale points) and decreasing the duration of the war (0.27), despite greater escalation potential (0.3, all with $p$ < 0.001). Patterns go in the same direction, but are substantively weaker and only partly significant for tank delivery. These results are tentatively in line with expectations H7 and H8 (for details, see Supplementary Discussion). Note that we observe no statistically significant differences in perceived consequences for domestic welfare, destruction, or

human suffering in Ukraine, or refugee flows by military aid types. Compared to air defense support only, ground troops are perceived to bring much higher expected escalation potential (0.78), together with economic disadvantages (0.38), but not suffering in Ukraine or refugees, while ground troops are also perceived to increase the chances of winning for Ukraine (0.28, all $p$ < 0.001) and decrease the duration of war (0.12, $p$ = 0.033). Note that compared to providing robust military aid (including combat aircraft), ground troops are not perceived to increase the chances of winning or end the war faster, while they are perceived to carry a much greater potential for escalation and domestic economic disadvantages ($p$ < 0.001). Hence, from the respondents' perspective, troops do not seem to add value relative to the delivery of offensive weapons. These patterns of perceived consequences go hand in hand with the patterns of support from Fig. 6: higher agreement with offensive military aid compared to air defense only is associated with perceptions of better outcomes for Ukraine, while respondents are seemingly willing to accept a moderately increased perceived escalation potential and see no negative side effects domestically. Low support for ground troops goes hand in hand with only moderately increased perceptions of positive outcomes for Ukraine, but substantially increased perceptions of escalation potential and domestic economic disadvantages.

The perceived consequences of doubling economic aid and maximally increasing sanctions against Russia are very comparable: these are seen as statistically significantly increasing winning chances (coefficients for doubling economic aid/maximal sanctions are 0.31/0.21, $p$ < 0.001) and shortening war duration (0.13, $p$ = 0.018/0.25, $p$ < 0.001); bearing no statistically significantly different additional escalation risk; statistically significantly easing suffering (both 0.21, $p$ < 0.001); decreasing refugee pressure (significantly so for economic aid, −0.18, $p$ = 0.001); but having relevant economic costs (0.30/0.33, $p$ < 0.001) (with air defense only as reference). Given that we see similar perceptions between economic aid and sanctions even for domestic economic costs, an explanation for the strong agreement to ramp up sanctions relative to doubling economic aid (see Fig. 6) lies outside of our inquired perceptions of consequences.

### Respondents with pro- vs. anti-Western attitudes show opposing perceptions of the consequences of economic and military aid

Next, we investigate, exploratorily, differences between subgroups with pro- vs. anti-Western attitudes as a central correlate for

**Fig. 7 | The consequences of different types of aid to Ukraine for Ukraine war outcomes and the domestic economy go hand in hand with observed patterns of support for military aid.** Coefficients for changes in perceptions of six types of consequences (as indicated in panel header) for six different types of Ukraine support (based on the split-sample aid vignette) relative to vignette level "air defense" as baseline. Error bars denote 95% confidence intervals from robust standard errors. $N = 10{,}011$ respondents. The estimates presented are based on separate linear regressions for each dependent variable on vignette condition indicators. Supplementary Table 15 presents corresponding statistics.

preferences over strategies for Ukraine. The vignette experiment allows us to investigate, first, whether these groups show—as the conjoint results let us expect—divergent agreement to the provision of aid types by their governments. More importantly, second, we can assess whether stated perceptions of the consequences of aid are similar—given respondents are exposed to the same information environment—or differ—given respondents might subjectively weigh and assess available information.

Regarding agreement with aid provision, we turn to absolute levels of agreement with aid types in our two respondent subgroups as presented in Supplementary Fig. 9. Interestingly, irrespective of underlying attitudes, all respondents are in (tentative) agreement with the provision of air defense (at around 4 scale points). However, we see a substantively strong divergence for respondents with strong pro- vs. strong anti-Western attitudes regarding more offensive military aid: compared to air defense only, provision of tanks and combat aircraft increases support strongly in the former (to 5.2), but decreases support in the latter (to 2.9). All groups clearly disagree with the provision of ground troops (3.2/2.3). Preferences also diverge regarding economic aid and sanctions, which are agreed with in the former, but disagreed with in the latter group. Overall, respondents with strong pro-Western attitudes assign ratings at or above four for any measure besides the deployment of troops, with the highest support for comprehensive military aid (5.2 scale points) and maximum sanctions (5.8 scale points). Those with strong anti-Western attitudes clearly disagree with all aid types, the more robust, the stronger, and only air defense is proximate to (tentative) agreement with 3.7 scale points. Taken together, this indicates that a subgroup of relevant size within Western countries is in strong opposition to aid packages that are required for a strong resolve against the Russian aggression.

Next, we investigate whether the differences in agreement with aid types between respondents with strong pro-/anti-Western attitudes as described above correlate with differing perceptions of the consequences of this aid. If not, both groups would, given a common information environment, at least have a common understanding of the 'ground game', even if they aspire to different states of the world. If yes, this would indicate that these groups not only differ in the weight they attach to a free Ukraine in their political preferences (i.e., their baseline resolve against Russia) but also in their assessment of the military effectiveness, international consequences, and domestic opportunity costs of aid types in the first place. This would imply that they differ in their perception of the extent of resolve Western support can deliver in the first place. Figure 8 presents results for the quartile with the strongest pro-Western and the strongest anti-Western attitudes. Estimates for the intermediate groups, whose preferences are situated between, are depicted in Supplementary Fig. 10.

Again, we conduct two types of comparisons: first, we compare absolute levels of perceived consequences of any additional aid to Ukraine among respondents with strong anti-Western (red diamonds) and strong pro-Western attitudes (blue circles). We observe that respondents with strong anti-Western attitudes do not relate any additional aid to increased winning chances for Ukraine or a quicker end to the war—but to higher escalation potential and more suffering with more refugees in Ukraine, as well as higher domestic economic disadvantages. Respondents with strong pro-Western attitudes relate any type of additional aid to higher winning chances, a quicker end to the war (besides air defense, air defense and tanks only, and doubling economic aid), no higher escalation potential (besides ground troops), and no increase in suffering/destruction (besides air defense and air defense and tanks only).

Second, we compare relative changes in assessments by experimental vignette conditions (see Supplementary Table 17 for statistics on the differences in marginal means). For respondents with strong anti-Western attitudes, we see no statistically significant evidence that different types of aid relate to differing consequences regarding Ukraine winning chances/a quicker end to the war, while, for respondents with strong pro-Western attitudes, these increase significantly ($p < 0.001$) and substantively with more robust military aid (including fighter jets, to 5.4/4.5 scale points) or ground troops (to 5.1/4.3), and also economic aid (to 5.0/4.1) or sanctions (4.9/4.3), compared to air defense only (4.2/3.8). In turn, for respondents with strong pro-Western attitudes, we find no statistically significant evidence that any type of aid (besides ground troops, to 4.8 scale points) increases perceived escalation potential, while this is the case for respondents with strong anti-Western attitudes

**Fig. 8 | Respondents with strong pro- vs. anti-Western attitudes differ in their assessment of the military effectiveness, international consequences, and domestic opportunity costs of aid types, i.e., they differ in their perception of the extent of resolve Western aid can deliver.** Marginal means for the perception of consequences (as indicated in panel header) for six different types of military and economic aid to Ukraine based on the split-sample aid vignette by subgroups of the first and fourth quartile of pro-/anti-Western attitudes. Error bars denote 95% confidence intervals from robust standard errors. $N = 4676$ respondents overall, with $N = 2367$ for the subgroup with strong anti- and $N = 2309$ with strong pro-Western attitudes. Estimates presented are predictions for consequences based on separate linear regressions for each dependent variable on the vignette condition indicators within subgroups. Supplementary Table 16 presents corresponding statistics, Supplementary Table 17 displays differences in perceived consequences with air defense as baseline.

for robust military aid including combat aircraft delivery (to 4.8, $p = 0.012$) and ground troops (to 5.2, $p < 0.001$), compared to air defense only (4.5). Regarding Ukrainian suffering or destruction, more refugee flows, and domestic disadvantages, we see that more robust military aid including combat aircraft (compared to air defense only) is related to increased perceptions of said consequences for respondents with anti-Western attitudes, but decreased perceptions for respondents with pro-Western attitudes. These patterns are statistically significant for suffering and destruction (with, for anti/pro-Western subgroups, $p = 0.003/ < 0.001$), and, for pro-Western subgroups, also for refugees ($p = 0.124/ < 0.001$) and economic disadvantages ($p = 0.165/0.008$). Also, respondents with pro-Western attitudes perceive statistically significantly less suffering or destruction with Western soldiers ($p = 0.007$) but also doubling economic aid ($p < 0.001$) or sanctions ($p < 0.001$) compared to air defense only. Respondents with anti-Western attitudes perceive statistically significantly more suffering/destruction, refugees, and economic disadvantages with ground troops ($p < 0.001$), and more economic disadvantages with doubling economic aid and sanctions ($p < 0.001$) compared to air defense only. Overall, pro- vs. anti-Western attitudes are clearly related to differing perceptions of how changes in military or economic aid affect the ground game in Ukraine, potential escalation, and the domestic economy.

In sum, the vignette experiment demonstrates that pro- and anti-Western segments differ in their views on the expected consequences of Western aid policies. Respondents with anti-Western attitudes not only express opposition against aid, and especially robust military aid—potentially prohibiting their governments from displaying resolve under circumstances of close electoral competition. Also, the concept of resolve, forcing opponents to back down based on the belief that strong military aid can turn the tide on the battlefield, does not resonate with this subset. The opposite picture emerges for respondents with strong pro-Western attitudes.

## Discussion

Regularly, states take sides, declare and reassure assistance, and provide actual support to third countries involved in interstate and intrastate crises and conflicts. These efforts intend to deter, and in case of the failure of effective deterrence, to fend off an opposing war party. Prominent cases include, e.g., US military aid to Taiwan, South Korea, or Israel. But rarely has such support been as visible and politicized as with the assistance to Ukraine defending itself against the Russian aggression since 2022. Historically, our case resembles the late 1930s, with Nazi Germany and other fascist countries menacing the first wave of democratization, when major Western societies likewise had to define appropriate strategies to deter and avoid destabilization and the spread of war. Berinsky[62] emphasizes how, then, the stance of the US public was essential for its president to maintain US resolve against Nazi Germany.

We propose that effective deterrence requires the credibility of government-to-government signals, i.e, in our case, the credibility that aid to Ukraine will be sustained[17]. Prior literature argued that this can favor democracies over autocracies because of audience costs. But game-theoretic accounts of democratic resolve (implicitly) take a unitary actor perspective for this expectation. This view ignores domestic heterogeneity and, thereby, potential democratic vulnerability from internal division. We argue that high audience costs from public commitments in democracies arise only in the highly stylized case in which democracies are united. But the nature of democracies is constituted by competitive elections. As soon as competing positions are campaigned on and citizens' resonances are heterogeneous, the credibility of statements by an elected government depends on the extent of the support for their positions in the electorate. The much-cited strength of domestically weak and fragmented systems in the context of international negotiations[63] can turn into a real weakness in threat games. Against this backdrop, we investigate Western citizens' reactions towards Ukraine support strategies of their governments in order to assess constraints to governmental resolve from observable, and publicly discussed, side effects of military and economic aid—showcasing barriers to withstand increasing autocratic challenges to the liberal world order[13].

Based on the literature on support for war, we assess three competing and intertwined considerations regarding support strategies: In

a normative orientation, citizens weigh principles of international law (primarily Ukraine's right to self-defense) against the question of which levels of death and destruction caused by sustained fighting are proportionate to this goal. In a strategic orientation, they are confronted with the risk of the war escalating beyond Ukraine versus the necessity of resolve against the Russian challenge to the liberal world order. In considering the economic implications, they face a relevant financial burden in supporting Ukraine, as well as the resulting domestic opportunity costs. Our results are based on conjoint and vignette survey experiments in the publics of the top 5 Ukraine aid-supplying Western countries ($N = 10{,}011$), the US, the UK, Germany, France, and Italy. We emphasize four core findings of our study.

First, results from our conjoint experiment indicate that, on average, citizens show strong support for the political sovereignty and territorial integrity of Ukraine, which we interpret as a credible signal of resolve against the Russian aggression. At the same time, citizens are very sensitive to human death and destruction among the Ukrainian, but not the Russian side, caring clearly about both Ukrainian combatant and civilian life. Similarly, fear of nuclear escalation is of high importance. However, concern about economic burden seems practically irrelevant. Although citizens are ceteris paribus cost averse, political strategies that entail high (0.6% of GDP) over low (0.2% of GDP) monetary costs are not strongly rejected. Insofar, we demonstrate that concerns over the economic costs of support are not at the forefront of citizens' reasoning, while strategic and human costs feature prominently.

Second, we show that this preference structure is surprisingly similar in the five countries under investigation. Based on a high-quality design, we are confident that this generalizes to actual population preferences. Concretely, the five country samples show an identical direction of preferences for all dimensions. A notable exception is a moderately differing spread in choice probabilities between the countries for the attributes on concessions and sovereignty (statistically significant between the UK and Italy, especially). This also corroborates earlier findings of Thomson et al.[59] and Politico[64]. Interestingly, the particularly pronounced resolve of the British public (also in comparison to the Italian public) also reflects in surveys conducted shortly after the start of the war (see, e.g., YouGov on February 28, 2022[65]).

Third, in an additional vignette experiment, we show that respondents' agreement with the delivery of concrete military aid increases the more robust this aid is. Respondents relate higher winning chances and a quicker end to the war with such aid; agreement is higher despite a likewise higher perceived escalation potential. This also indicates considerable resolve by average citizens.

These three findings shed light on recent theoretical and practical debates: Regarding theory, our case is in line with the argument that citizens take moral considerations extensively into account when forming foreign policy preferences[47]. We extend this literature, assessing the case of an ally's confrontation with a nuclear power, in a war on foreign soil invaded by foreign troops, where human suffering is spatially very distant (not involving killing of or by domestic soldiers). Still, in our case, human life abroad seems to be valued much higher by Western citizens compared to domestic economic costs, at least regarding life among allied parties. Although other research has emphasized materialistic preferences and economic calculations as highly relevant for the formation of foreign policy preferences (see, e.g., refs. 66–68, but ref. 69), this is not the case here. In addition, for our case, the strategic risk of nuclear escalation is plausible. While weighted strongly by the respondents, escalation risk does not crowd out the relevance of the moral trade-offs we studied. Future research could investigate more complex perceptions related to the risk of nuclear escalation, e.g., genuine strategic concerns (that the war extends to the domestic realm), or secondary humanitarian considerations (related to the fallout of nuclear strikes). Moreover, our

case reflects recent arguments about the relevance of military and civilian casualties in the war-support literature. For example, Dill & Schubiger[26] show that US citizens despise (foreign) civilian death—in line with principles of international law. Notably, they barely care about enemy military casualties. Most relevant is, however, not endangering domestic military life. Our results reflect the former two results in the context of a war fought on foreign soil. In addition, our findings link to the latter result, reflecting strong care for the life of allied military personnel, on par with Ukrainian civilian life—and much different from respondents' assessment of Russian military casualties, which, in some subgroups, are even favored. Last, our study extends findings from the war support literature, with its heavy US focus [for exceptions, see, e.g., ref. 29], to populations of four NATO middle powers, which are practically highly relevant for Ukraine aid and security policy beyond (e.g., with France and the UK as UN security council members, or all four as among the largest democratic arms exporting countries next to the US). Future research would need to assess the extent to which our findings generalize to Eastern European countries that share a border with Ukraine and/or Russia.

Regarding policy, our results indicate, on the one hand, that when Western countries decide on support strategies, they have to take into consideration that popular support is not unconditional, i.e., that (popular) resolve has specific constraints. Given that foreign intelligence is likely both informed about these constraints and might even, in the age of social media, try to influence them through stealth misinformation campaigns[70], this emphasizes the importance of the style of communication between governments and their population on these matters. The credibility and content of elite cues[71], how united elites attempt to educate and rally their publics, then becomes pivotal. Moreover, our study indicates that the boundaries that citizens draw on Ukrainian support seem to differ relevantly from their governments, constraining the actions of the latter. For example, minimizing Ukrainian casualties should be a top priority for Western governments if they want to maintain a strong resolve. Preventing nuclear escalation at the same time seems of utmost importance. Here, counter-communication against autocratic signaling that strategically manipulates perceived risks of nuclear escalation seems to be crucial for sustained support[72]. Given that the monetary costs of aid, particularly military aid, do not appear to be a relevant concern, Western governments should not be constrained by public opinion in increasing military aid significantly. This also indicates that party-level contestation, e.g., in the US surrounding domestic economic opportunity costs, does not resonate similarly in public sentiment (see related findings of Rudolph[73]). Future research could investigate whether alternative operationalizations of our experimental cost cue could play a role here[74], i.e., whether similarly plausible depictions of economic costs (e.g., on a household or taxpayer level) yield similar results. On the other hand, our results show a relevantly divergent preference pattern compared to studies of Dill et al.[31,32] on the preferences of the Ukrainian population. We can directly contrast with their results, since we fielded several attributes they inquired about. They find almost unconditional support in favor of defending territory and sovereignty against Russia among the Ukrainian population (despite human costs and the risk of escalation) in 2022[31], which persists strongly, particularly regarding political sovereignty, in 2024[32]. That the populations of the Western support coalition apparently give greater weight to human costs abroad than the affected population seems surprising. One potential explanation is that only direct involvement can confer moral legitimacy on this very hard trade-off. But at the same time, the Ukrainian population is also more strongly aware of the human costs of giving in to Russian aggression, a factor external to these experiments. Regarding potential agreements for a freezing of the war or even a lasting peace, the Ukrainian population rejects territorial concessions of only the Crimean peninsula nearly as strongly as concessions as of the frontline in late 2022 to 2025 (relative to full integrity)—among

Western populations, this differs by more than a factor of two. This indicates a potential conflict between potential peace deals that could seem acceptable to Western populations compared to Ukrainian citizens.

Turning to our fourth core result, our evidence indicates important heterogeneity within Western democracies. The fundamental dividing factor on the extent of resolve is actually citizens' attitudes towards NATO, the US, and Russia, which we interpret as the extent to which they support the current liberal world order. Among respondents in the quartile of strongest pro-Western attitudes, support for Ukraine is almost unconditional and is undertaken at high economic, strategic, or human costs. Among the quartile of strongest anti-Western attitudes (i.e., at least 25% of citizens), respondents do not care for Ukraine's survival as a sovereign political entity, but despise higher economic, human, and strategic costs. The former agree with all types of aid to Ukraine and increase agreement with more robust military aid, the latter show inverse preferences. In particular, the former connect more robust aid to better outcomes for Ukraine on the battlefield, the latter do not, a potential indication that they do not perceive that military aid can actually deter Russia.

These findings indicate a divide in foreign policy preferences within Western democracies that is counter to traditional left-right dichotomies. This gives the potential for political coalitions to form beyond traditional cleavages. With regard to Ukraine policy, we can expect the 25% of citizens each in the first and fourth quartile to show rather stable attitudes (given the correlation of attitude extremity and attitude strength[75]) while those in the middle of the preference distribution decide on the political majority. At the time of our survey, the respondents in the second and third quartiles tilted towards the pro-Ukraine camp, but as elite cues can shape public opinion[71], the consistency of resolve that democracies can show becomes part of a political game. Future research could study in particular whether (some) political elites cater, e.g., due to short-term office-seeking motivations, to the anti-Ukraine camp. Importantly, as we fielded identical surveys among the populations of the most prominent supporters of Ukraine (in absolute terms) at the same point in time, we can show that differences in government behavior likely stem from differences in institutional structures that translate citizen preferences into representation and not from differences in voters' majority preferences. For example, the minimum winning coalition[76] in presidential and majoritarian electoral systems like the US or UK is substantially lower (about 25%, i.e., half the population in half the constituencies/ electoral college to win) compared to systems with proportional electoral systems like Germany or Italy (about 50%). Given that we find a sizable but minority share of citizens indifferent to the fate of Ukraine in all five countries, institutional structures that determine how strongly such subsets shape government composition can become highly influential for government position-taking regarding Ukraine aid specifically and resolve against autocracies more generally. Future research could explore whether the change in US policy between the Biden and Trump administrations is related to differences in the preference structure of their respective core supporters, as our survey indicates. Lastly, our results indicate a substantive difference in the way subpopulations assess factual information. Again, future research could investigate how, based on these different perceptions, new information and (partisan) elite cues are taken up. More generally, it would be fruitful to connect our research to questions of issue saliency, attitude stability, and preference change in the future.

We conclude by highlighting the strengths and limitations of our survey-experimental approach. The survey-experimental approach we apply allows us to causally relate experimenter-induced variation in question display to differences in response behavior[35,77]. This improves on a number of known concerns from traditional survey measures for assessing policy preferences[78], especially when inquiring about aspects that each have valence dimensions (e.g., lower economic, human, or

strategic costs). The vignette experiment on aid types has the core advantage of linking respondent replies to one (experimentally revealed) vignette each. This precludes spillovers from respondents' assessments of one aid type onto others; i.e., we can causally relate differences in agreement and differences in consequence perceptions to aid types. However, we cannot causally separate whether stated perceptions of consequences are causally prior to stated agreement to aid or not. That is, our design cannot unravel whether respondents agree with certain types of aid because of perceived consequences or whether they adapt their perceptions of consequences to what they agree with (e.g., via motivated reasoning). On the one hand, Berinsky[79] convincingly argues that aspects such as expected military death, the existence of a rightful cause, or perceived success probabilities causally determine support for military action; but on the other hand, it is also probable that underlying respondent characteristics (like partisan predispositions) jointly affect the perceived consequences of aid types and agreement with the aid itself. To establish causal direction, additional experiments would be required[80]. We deem it still insightful to infer the overall structure of agreement and perceptions of more vs. less robust types of aid, as debated at the time of our survey, given that these are likely to directly relate to domestic pressures on governments. The conjoint experimental design allows us to assess, with a causal interpretation, the relative importance of our attributes at the time of the survey, as all attributes must be considered simultaneously by respondents in the choice task. Given our monitoring of public debates at that time, we are confident that we inquired about its most salient aspects, which, thereby, can also be realistically assessed by respondents. However, an experiment is always a stylized depiction of reality. For example, neither we did spell out civilian suffering in the occupied Russian territories, the risk of a Russian invasion of NATO territory. If respondents read aspects of unmentioned attributes that could co-occur with mentioned attributes in the task, this could mask preferences[81]. Given our nine-attribute design, we deem it unlikely that this could substantively alter our findings, however. A relevant second concern is the hypotheticality of decision-making in such experiments, given that drastic real-world changes of attributes are usually more salient and take place in a more complex information environment. At the same time, previous research indicates that survey-experimental decision-making can mirror real-world voting[35] and preference formation[82], and that swings in public opinion based on conjoint experimental conditions can represent swings based on real-world changes [see ref. 25, studying weapon transfers, relating to Ukraine]. Third, as surveys elicit stated preferences rather than actual (voting) behavior, they can be prone to biases, e.g., social desirability bias. However, factorial survey experiments, as our conjoint, exhibit the desirable property of minimizing social desirability bias considerably[83]. Fourth, resolve against autocratic aggression will, in the end, depend on governments' actions. However, as we argue, understanding democratic governments' behavior goes hand in hand with understanding the preferences of their citizenry. With this argument, we build upon a growing scholarship that indicates that foreign policies attract citizens' attention in a similar way as domestic policy, i.e., that citizens form preferences consistently in both domains[47,52,84], and that citizens' opinions, as measured in surveys, seemingly matter for policy debates, political mandates[85–88], and ultimately for policy processes[89,90] and elite decision making[91–95].

## Methods

Our study, including both experiments, has been approved by the Research Ethics Board of the Faculty of Social Sciences, LMU Munich (Study Protocol GZ23-02). Participants gave their informed consent and participated voluntarily. We conform to all applicable data protection regulations. Participants in the survey were compensated by the survey company YouGov. The study was pre-registered, and the pre-registration is available at Open Science Foundation under

accession code https://doi.org/10.17605/OSF.IO/TVZSA, also including the survey instrument.

## Sampling strategy

Overall, our goal was to sample 10,000 respondents. We chose this large sample size to provide sufficient statistical power to detect small effect sizes with reasonable confidence (e.g., for $N = 10,000$, $Alpha = 0.05$, and $Power = 0.8$, the minimum detectable AMCE is 0.014, see pre-registration). For data gathering, we contracted YouGov, drawing on a high-quality quota sample from YouGov's online access panels in the US, the UK, Germany, France, and Italy. These countries are the top 5 providers of combined bi- and multilateral military and economic aid to Ukraine [ref. 6, p. 29], and are central to decision-making in both NATO and the European Union (EU). The survey field time was from June 14 to August 28, 2023, with almost 90% ($N = 8956$) of responses completed by July 1.

We used voting eligibility as the core inclusion criterion (i.e., we survey adult resident citizens, which includes Irish citizens and citizens of a qualifying Commonwealth country for the UK). YouGov has automated panel-level systems for detecting straight-liners and persistent speeders, which were excluded before data delivery. Survey participants were selected from the YouGov panel using national-level demographic and political quotas: region, gender, education, age, past voting behavior, political interest (used for UK, French, Italian, and German samples), and country-specific population characteristics (race and home ownership for the US; urban/rural for France and Germany; the 2019 EU vote for Italy; work sector for France; social grade and the EU referendum vote for the UK). In all countries, YouGov uses specific, proprietary interlocks. We thereby aim at the actual distribution of general political preferences across countries. YouGov benchmarks this against ongoing political opinion polling and validates it against official election results. Their scheme produced vote intentions that accurately predicted electoral outcomes in the run-up to the last national elections in each of the five countries.

YouGov completed 12,009 replies across all 5 countries. To match the population of the countries as closely as possible, YouGov retained a set of quota-optimized ~2000 respondents per country from the original 12,009 responses. This gives us an $N$ of 10,011 as the final sample on which we rely. Note that YouGov also provided weights to achieve the set quotas even more effectively. These weights (max. 6.83 ($N = 1$), min. 0.39 ($N = 2$)) are based on an iterative proportional fitting algorithm, and we use these as survey weights for all analyses.

This setup was agreed upon with YouGov prior to data collection and is part of our pre-registration (see Supplementary Discussion section 1.2). We are confident that we can improve on most current research based on quota-based sampling strategies. Current work mainly draws on quoting based solely on socio-demographic characteristics or even convenience sampling without proper validation, and seemingly struggles to achieve population representativity[96]. Our approach explicitly aims to obtain a sample representative of the current distribution of political views in each country, thereby enabling the generalizability of our findings to citizen preferences in those countries.

We present descriptives for our sample in Supplementary Information Table 18. Note that self-reported gender identity (male or female) was used as a quota for survey data collection to ensure representation by demographic groups within each country. Gender was interlocked with age (for France), with age and education (Germany, UK, Italy), and with age, education, and race (USA). The reported findings apply to all sexes. Disaggregated gender data are provided in the individual-level dataset contained in our replication files[97]. Since the focus of this study is not to investigate the effect of socio-demographics on attitudes, no (post-hoc) analyses of gender differences are included.

## General survey design

Our survey had a median duration of 20.08 min. Participants first responded to questions about their socio-demographics and general political viewpoints (used for quotas). Next, respondents completed a conjoint experiment on general arms trade profiles, followed by attitudinal questions on arms trade in general (reported in ref. 98) as part of a larger project. The respondents then participated in the Ukraine strategy conjoint experiment and the subsequent aid vignette experiment, which are at the core of this article. Subsequently, we inquired about foreign policy values, attitudes towards war and peace, and pro-Western attitudes. The former two were administered to a random subset of respondents (50%) to shorten the overall survey duration.

The display of levels in both the conjoint and the vignette experiments was uniformly randomized; respondents were blinded and debriefed at the end of the survey. Given concerns about bias from attribute-order effects[35], we follow the recommendation of Rudolph et al.[99] and use block-randomized attribute ordering within dimensions. This cautions against attribute-order effects while facilitating respondents' comprehension of the choice tasks. The outcome variables were measured using choice or Likert scales, with the scale direction randomized in almost all cases. The exact survey flow and question wordings are provided in the master questionnaire attached to the pre-registration. Note that the authors designed the questionnaire in German and English. The French and Italian versions were translated by professionals (commissioned by YouGov). All versions were proofread by native-language social scientists.

## Set-up of the conjoint experiment

Our first experiment is a paired-profile conjoint experiment over four rounds, with a choice and rating task, each comparing two strategies to end the war in Ukraine. This yielded $N = 10,011 * 4$ rounds $* 2$ profiles $= 80,088$ replies. In total, we confront respondents with nine attributes (which cluster into four dimensions of theoretical interest) with three to, in one case, four levels. Columns three and four of Fig. 1 provide an overview of the attributes and their levels; the survey wording as presented to the respondents (in the English version) is presented in Supplementary Table 1. The attributes were partly inspired by Dill et al.[31], who examined war-strategy preferences in a Ukrainian sample. We adapted their design to our research question and temporal context. In comparison, we expanded the human cost dimension to fielding attributes on Russian military casualties and Ukrainian infrastructure, added the economic cost dimension with two attributes, and updated levels to our temporal context.

## Attribute and attribute level selection in the conjoint experiment

The justification for the attributes fielded in our four dimensions was as follows (see also pre-registration): Human costs of war relate to, on the one hand, the military dimension, i.e., military casualties. Given that public debate regularly differentiates Russian and Ukrainian casualties [see, e.g., ref. 100], we depict two separate attributes on the prospective number of Russian and Ukrainian soldiers killed. This allows us not only to trace preferences over the human costs of war (in the military realm) but also to examine differences in which side of the conflicting parties bears these casualties. On the other hand, we consider human suffering along the civilian dimension, i.e., civilian death and destruction. Here, we consider both direct human suffering, the number of Ukrainian civilians killed, and the amount of infrastructure destroyed in Ukraine. The nuanced depiction of this dimension allows us to separate direct (death) and indirect (infrastructure) costs, death among civilians and the military, and casualties on the side of the aggressor and the defender.

Financial costs of aid for the countries of the support coalition, we capture with two attributes: military and economic aid, given the

prominence of domestic opportunity costs in public debate [for respective Ukraine strategy arguments, see, e.g., refs. 101,102] and literature. The two attributes in this dimension also allow us to gauge the trade-off between the economic and military aid domains.

Strategic risks for a serious threat of escalation are operationalized via the likelihood of employment of nuclear weapons in Ukraine[103], which looms large over public debates on risks associated with the war [see e.g., ref. 104].

Political costs of Ukraine strategies, and, in turn, containment of the Russian regression, mainly pertain to "the boundary between the two countries and the degree of freedom of Ukraine to join NATO and the EU", as also proposed by D'Anieri [ref. 105, p. 319]. Concerning the former, central demarcation lines of a potential peace agreement surround the status of 2014 annexed Crimea (of stated pivotal importance to Russia), the status of the Luhansk and Donetsk Oblasts occupied by Russian troops and allies since 2014, and the status of territories in eastern Ukraine occupied by Russian troops since 2022. Concerning the latter, central demarcation lines concern an EU or even NATO membership of Ukraine, voiced as an existential threat to Russian geo-strategic interests from the Russian perspective—"Russian leaders made it perfectly clear that bringing Ukraine into NATO would be crossing 'the brightest of red lines'"[103]—as well as Ukraine's ability to choose its own mode of economic and political organization, i.e., the extent of influence of Russian over its government [ref. 105, p. 319f.].

The levels of the above-mentioned attributes were chosen based on linear projections of publicly available data on the war's consequences in its first year. This information is, in principle, also available to all respondents as it constitutes the general news environment. Importantly, we communicated to respondents that we highlight "The expected effects from today until the end of the war [... while] [n] umerical values are given as forecasts over the next 12 months" (see questionnaire in the pre-registration). The detailed rationale for attribute levels was as follows:

For the presented Russian military casualties, we drew on estimates by the UK Ministry of Defence (March 27, 2023), which estimates the number of Russian soldiers killed since the war's onset at 40–60,000 troops[106], and we take the mean as our medium level. We used linear extrapolation for the medium level (50% for the lower and 200% for the upper level; see also ref. 107). For the numbers of civilian casualties shown, we used the UN OHCHR count of civilian casualties (as of 20 March 2023)[108] in the Ukraine-Russian war, stating 8317 killed civilians. These were rounded to 8000 with linear extrapolation for the medium level (50% for the lower level and 200% for the upper level). We explicitly do not differentiate between civilian casualties in the territories held by the Ukrainian vs. Russian government, as, on the one hand, we wanted to reduce the number of attributes respondents have to process, and, on the other hand, we theoretically do not expect that respondents differentiate between where civilians are killed. Regarding infrastructure damage levels, we used estimates from a UNDP report[109,110], which calculated infrastructure damage since the beginning of the war to $100B USD (as of 16 March 2023). We used this as a realistic medium level, and 50% (100%) as lower (upper) levels.

For the stated amounts of military and economic aid for each country, we used data for military, humanitarian, and financial aid to Ukraine since the beginning of the war (as of April 01, 2023) by Trebesch et al.[6]. We used total bilateral commitment for the UK and the US, and total bilateral commitment, as well as the countries' shares in EU commitments, for EU countries. This compilation leads to the past support levels shown in Supplementary Table 2. From this, we take the median level of GDP support (0.36% of GDP), allocate it evenly to potential future military and civilian aid (0.18% of GDP each), and round it to 0.2% for easy-to-comprehend values. This constitutes the medium level for this attribute. Extrapolating to 50% and 150%, we obtain the lower and upper limits for this attribute (0.1% and 0.3%). We communicate this amount both in absolute terms and relative to each country's GDP, so

respondents can understand the actual size of aid while we, at the same time, set a comparable relative baseline across country samples. The mid-level of this attribute is well correlated with the actual total compound aid flows as tracked by Trebesch et al.[6]. Our calculations are detailed in Supplementary Table 3. We intentionally communicate costs in terms of GDP rather than presenting the calculated tax burden at the individual or household level, as this aligns with public debate and news coverage across all five countries. Deriving potential, but realistic, individual/household costs would depend on tax law configurations, which vary widely across countries and within samples and are interpreted differently by individuals along the income scale. Since political decisions are usually made and communicated by interpreting costs as a percentage of GDP rather than their impact on individual pocketbooks, this depiction matches our theoretical starting points. Note that this operationalization is firstly realistic in the sense of taking sensible starting values for overall support in each country; secondly comparable, as support by GDP levels has the same levels in all countries; thirdly informative over policy both with respect to overall support (ranging from 50% with low/low to 150% of the initial war years' aid with high/high attribute level expressions); fourthly flexible in the allocation between military and civilian aid, allowing us to assess trade-offs between both. If we had taken actual military and actual civilian aid levels as starting points, we could only have reproduced policy as currently conducted by countries in the conjoint (e.g., the US gives relatively more military compared to civilian aid, i.e., citizens would have had to choose between profiles where military aid is always larger than civilian aid). Also, note that we did not adjust numerical values by exchange rate, as exchange rates were reasonably close to parity (as of March 1, 2023: 1 Euro = 0.9 Pounds; 1 Euro = 1.06 USD), and adjustment would have led to more complicated depictions of numerical values for US and UK respondents.

For the stated nuclear strike risks, we follow Dill et al.[31] in proposing that the risk of a Nuclear strike by Russia on Ukrainian territory can be bounded by 0%, 5%, and 10%. The latter is the upper limit following expert assessments of the military situation on the ground (see, for a detailed justification, ref. 31, p. 6).

Regarding political costs, the design of the different levels showing territorial cessions is based on the following rationale. As a first realistic scenario for cessions, we set the line of contact between Russian and Ukrainian military forces (as of April 01, 2023, with only marginal movement until 2025)—thereby, Russia would gain about 16% of Ukrainian territory in eastern Ukraine. We see this as a realistic upper limit on cessions, given public debates on a "freezing" of the war at the current frontline (see, e.g., refs. 111,112). A second realistic scenario would be the since 2014 controlled areas of Russia-allied rebel groups in the Luhansk and Donetsk region, as well as the Crimean peninsula—thereby, Russia would gain about 8% of Ukrainian territory in Eastern Ukraine. A third realistic scenario would be the since 2014 annexed Crimean peninsula—thereby, Russia would gain about 4% of Ukrainian territory in Eastern Ukraine. A fourth realistic scenario would be no territorial cessions (i.e., 0% of Ukrainian territory). Given that respondents likely have no clear understanding of the extent of the implied concessions, we calculated and presented the respective landmass loss for a comparable presentation. Again, this operationalization is, with slight adaptations, similar to Dill et al.[31]. For levels of Ukrainian sovereignty, we take the Russian war goals (Russia-controlled government in Kyiv) and the Ukrainian war goals (full sovereignty with the possibility to join EU/NATO) as extreme points. A possible mid-scenario is a neutral status (self-determination without the possibility of joining EU/NATO). This operationalization is, with slightly adapted wording, similar to Dill et al.[31].

## Set-up of the vignette experiment

Subsequent to the conjoint experiment, we presented respondents with one of six different scenarios of concrete aid to Ukraine (see columns two and three of Fig. 5): four scenarios of military aid with

varying offensive potential and two scenarios of economic support (increased financial aid to Ukraine or economic sanctions on Russia). Hence, we field a 1 × 6 split-sample vignette. The vignette was followed first by a question on general agreement with respondents' governments providing such aid and a battery of questions on perceived consequences of aid provision for Ukraine and the domestic economy.

Military aid scenarios relate to ongoing debates during the time of the survey on types of weapon systems potentially to be supplied to Ukraine: air defense (as decided among NATO countries to be delivered since late 2022[113]), additionally combat tanks (as discussed among NATO partners since January 2023, with a decision in favor of deliveries ultimately taken[114]), or additionally combat aircraft (as discussed among NATO partners since May 2023, with no decision taken as of June 2023[115]). Military support also relates to the employment of NATO ground forces, ruled out by NATO countries as of June 2023, even though debated at times[116,117]. Regarding the dimension of economic aid, we presented a doubling of the effort of the supporting coalition, as proposed in several countries at the time[118]. A final vignette proposed stronger economic sanctions against Russia, i.e., support outside the military realm, as regularly called for in public debate[119]. For all vignette presentations, we communicated a common statement of actual military and economic aid levels at the time of the survey, and proposed additional aid with our vignette. This generated a common baseline across respondents. We inquired (dis)agreement with aid provision on a 7-point scale. We also allowed for a "don't know" option, excluding these (around 200–400 respondents per country) from subsequent analyses.

Subsequently, we investigate the perceptions of respondents on how this would affect the ongoing war (war intensity, escalation potential, effectiveness of Ukraine, war duration) and the respondents' own country (economic consequences, refugee flows) along six dimensions: better chances for Ukraine to win the war against Russia; higher escalation potential, i.e., a higher risk of a third world war; a quicker end to the war; overall more suffering and destruction in Ukraine; more Ukrainian refugees from Ukraine; economic disadvantages for [country]. The respondents gave their replies on 7-point scales. The questionnaire in the pre-registration presents the exact wording of these dimensions.

## Non-experimental variables and measures

To assess subgroup heterogeneity, we rely on non-experimental items that capture respondents' underlying attitudes and values. At the core of our article is the assessment of heterogeneity in terms of how strongly citizens are rooted in the liberal world order, which we term pro- or anti-Western attitudes and measure by their extent of transatlanticism.

These are summarized by three items: first, a measure of anti-Americanism ("American imperialism is the real threat to world peace."–English translation of a German item used by Knappertsbusch [ref. 120, p. 207]); second, a measure of Russophile attitudes ("I can understand that Russia feels threatened by the West."–English translation of a German item asked in a survey of the polling company Infratest Dimap for the "ARD-DeutschlandTREND" in January 2022 [ref. 121, p. 9]); third, a measure of attitudes towards NATO (based on a survey in a post-election study by the German Marshall Fund of the United States on the 2004 US elections [ref. 122, 6 (Q4)], slightly reformulated). To construct quartiles of pro/anti-Western support, we conducted a Principal Component Analysis (PCA) using the above-mentioned items. Only the first principal component showed an Eigenvalue greater than 1 (Kaiser criterion), namely 1.62 (explaining 54% of the variation), which we used to predict each respondent's value on an anti-Western dimension (respondents who answered "Don't know" to the question on NATO were excluded). Respondents in the first (second) quartile (i.e., 25% of the sample) of this scale were classified as strongly (moderately) anti-Western, those in the third (fourth) quartile as moderately (strongly) anti-Western.

For additional (also pre-registered) subgroup analyses, we drew on general attitudes towards war and peace and foreign policy values. Regarding attitudes towards war and peace, we relied on the battery proposed by Bizumic et al.[123]. To reduce survey duration, we shortened this scale to six items based on inter-item correlations in the data of Rudolph et al.[25]. In line with the scale validation by Van der Linden et al.[60], our PCA of the battery's items yielded two dimensions (based on the Kaiser criterion of retaining components with Eigenvalues above 1; in our case, 2.57 and 1.07, explaining 62% of the overall variation). After factor rotation, we predicted the respondents' scores of these two components. The first component captures attitudes towards peace–this component loads (positively) particularly on the items "We must devote all our energy to securing peace throughout the world", "Any use of force between people is always and everywhere immoral", "There is no conceivable justification for war". The second component captures attitudes towards war and loads (positively) particularly on the items "In general, I am not too concerned about peace in the world", "War is sometimes the best way to solve a conflict", "Under some conditions, war is necessary to maintain justice". Next, we identified respondents in the top and bottom quartiles of the resulting scale to classify respondents with extreme values. We also use the PCA to operationalize pacifist respondents as the cross-over of the top quartile of attitudes towards peace and the bottom quartile of attitudes towards war, and vice versa for hawkish respondents (top quartile of attitudes towards war and bottom quartile of attitudes towards peace).

To measure foreign policy values, we relied on an item battery proposed by Kertzer et al.[47], designed to assess three dimensions of underlying attitudes. Again, for reasons of survey duration, we shortened this scale to five items based on inter-item correlations in the data of Rudolph et al.[25]. We derived the first three components (with Eigenvalues of 2.06, 1.00, 0.91) from a PCA, from which we predicted three indicators. We also use the third component, even though slightly below the Kaiser criterion, based on fit with theory[47]. After rotation, the first component loads (positively) on three items ("[country] needs to cooperate more with the United Nations", "Promoting and defending human rights in other countries is of utmost importance", "Helping to improve the standard of living in less developed countries is of utmost importance"), the second component on one item ("Going to war is unfortunate but sometimes the only solution to international problems"), the third component on one item ("We should not think so much in international terms but concentrate more on our own national problems"). These three components directly capture what Kertzer et al.[47] name "cooperative internationalism", "militant internationalism", and "isolationism". Again, we identify respondents in the top and bottom quartiles of the resulting scale and estimate separate marginal means for these respondents with extreme attitudes.

## Estimation strategy and robustness

For estimation, we rely on Average Marginal Component Effects (AMCEs, Hainmueller et al.[35]) and their interactions, and Marginal Means (MMs, Leeper et al.[56]) for subgroup comparisons. AMCEs are estimated using linear OLS regression. This implies conducting two-tailed tests for each estimate (also for the related marginal means[56]), including the calculation of corresponding $p$-values and confidence intervals. We cluster standard errors on the respondent level for the conjoint experimental analyses[35], and use robust standard errors for the vignette experimental analyses. In additional models, we calculate Bonferroni-corrected confidence intervals for the main article conjoint subgroup analyses.

As detailed in the preregistration, to assess the validity of our inferences, we checked the core assumptions of the conjoint experiments as outlined by Hainmueller et al.[35], namely that there are no carry-over effects between rounds and no profile-order effects. The

corresponding AMCE comparisons (see Supplementary Figs. 13 and 14) show no concern, which is why we continue to pool all data. Note also that, with respect to the assumption of full randomization of the attribute levels, we checked their distribution, which is uniform as expected. Since AMCEs are nonparametric tests and based on a fully randomized experimental design, testing of assumptions such as normality and equal variances is unnecessary. Finally, we engage in recent debates on AMCE interpretation regarding majority preferences[57,124] and subgroup preferences[56]. First, for subgroup differences, we follow the recommendation of Leeper et al.[56] and interpret marginal means, noting that comparisons with AMCEs would yield comparable interpretations. Second, we assess whether our interpretations are robust to Average Component Preferences (ACPs; Ganter[57]), which control for attribute-level ties, thereby directly enabling the comparison of preference patterns across attributes with varying numbers of levels. These are reported in Supplementary Tables 15 (overall) and 16 (pro-/anti-Western) and show the same pattern of results as our AMCE/MM plots. Note that ACPs are based on the unweighted sample, as the R package provided by Ganter[57] does not allow survey weights. Third, we examine potential preference cycles and assess potential artificial transitivity that could bias AMCEs[58]. Calculating AFCPs (see Supplementary Tables 5 and 6), our data show no indication of artificial transitivity.

### Reporting summary

Further information on the research design is available in the Nature Portfolio Reporting Summary linked to this article.

### Data availability

The raw data on which all analyses in this article are based are available from Harvard Dataverse under accession code https://doi.org/10.7910/DVN/UDBPS1.

### Code availability

Replication codes to reproduce all analyses conducted for this article are available from Harvard Dataverse under accession code https://doi.org/10.7910/DVN/UDBPS1.

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

## Acknowledgements

We thank audiences at the University of Mannheim, the University of Konstanz, the Berlin School of Economics and Law, the 14th Annual Conference of the European Political Science Association 2024, the 29th Academic Convention of the German Political Science Association (DVPW) 2024, Thomas Bräuninger, Bernd Schlipphak, Carsten Wegscheider, and Tobias Börger for feedback on our manuscript, and Carl Mueller-Crepon for feedback on the survey instrument. We acknowledge financial support by the German Foundation for Peace Research (DSF), grant no. FP 07/22FB1-PRO-07 (PIs: L.R. and P.T.), by the Heinz and Sibylle Laufer Foundation (FH), as well as open access funding by the publication fund of the University of Konstanz, project DEAL, and from LMU Munich.

## Author contributions

L.R. had the lead in conceptualizing the study, in conducting the analyses, and in drafting and revising the manuscript. F.H. led data collection and contributed to data analysis. P.T. contributed to argument development and drafting. P.T. and L.R. acquired funding. All authors contributed to the questionnaire development and manuscript preparation.

## Funding

## Competing interests

The authors declare no competing interests.
