## [Transparent Peer Review file · Nature Communications]

Examining public support for Ukraine's defense against autocratic aggression

Corresponding Author: Dr Lukas Rudolph

Version 0:

Reviewer comments:

Reviewer #1

(Remarks to the Author)

This paper employs a five-country sample to explore the impact of hypothetical changes in the military situation in Ukraine on expressed support for the Ukrainian effort. The authors employ a large-scale conjoint experiment to assess perceived support across a variety of human and economic costs, as well as ultimate strategic aims. They also use a vignette experiment to try to assess the mechanisms by which respondents assess the perceived consequences of action. The authors find that variation in the human costs of war on the Ukrainian side lead to different levels of support, as do various war aims (containment of aggression and sovereignty). Economic costs, however, are largely unrelated. The pattern of results is very similar across countries, indicating that while some differences may exist in the structure of the preferences of citizens of different nations, there is much more similarity across the U.S. and Western Europe.

This paper is first and foremost an empirical paper. The underlying mechanisms are drawn from other work and there is little new in terms of theoretic advance. Whether such a paper is worthy of publication in Nature Communication is a decision that the editor must make. Personally, I think that the advancement of knowledge through empirical work is a worth endeavor and my comments should be read as suggestions to improve the description and presentation of the results.

I think that this paper is interesting and tackles an important (and current) topic. I also appreciate the scope of the study – the authors ran five well-powered surveys to compare preferences across countries and the data collection itself is an impressive undertaking. However, I think that the authors make a stronger causal claim than is warranted by the design of the study. I suggest that the authors tone back their language in a few places and discuss the limitations of employing hypothetical scenarios in a real-world (and ongoing) conflict.

On page 5, the authors claim that their results are strongly identified because they draw on “conjoint and vignette experiments that allow the identification of causal effects.” This is very strong language. I would note that the entire experimental setup relies on individuals responding to hypothetical scenarios, not changes in real-world events. Some political scientists and economists would argue that such a design can only measure “cheap talk” hypothetical preferences. I am not one of them, but I do think the authors need to recognize and acknowledge the potential limitations of employing hypothetical scenarios. This piece by Susan Hyde in the Annual Review of Political Science (<https://doi.org/10.1146/annurev-polisci-020614-094854>) is somewhat old but could be a good place to start a discussion of the strengths and limitations of survey experiments.

On a related note, I found the vignette experiment interesting, but I was not convinced that the various dependent variables were strong tests of the mechanism at work. The perceived consequences of action (chance of Ukraine winning, quicker war ending) might be causally prior to overall support. But it could well be that the consequences are themselves endogenous to support levels. The exchange between Berinsky and Druckman on the one hand, and Gelpi and Reifler on the other concerning the causal impact of war success (in *Public Opinion Quarterly* – starting with, Vol. 71, No. 1, Spring 2007, pp. 126–141) could be helpful in reframing the strong causal claims made in the paper.

I also had a few specific notes:

1. Section 2.2 needs to be expanded. In particular, the discussion of the theoretic and empirical setup of the vignette experiment should be fleshed out. The links to the various theories discussed earlier in the paper were not clear and I had to read the section several times to understand exactly what the authors were doing (and even then, I was sometimes confused).

2. In the appendix, the authors make note of recent criticisms of the use of ACMEs to assess majority preferences and subgroup preferences. They find the pattern of results hold when using ACPs and AFCPS. I would mention this fact in the body of the paper to avoid confusion, given the salience of recent debates about this matter.

Reviewer #2

(Remarks to the Author)

This is an interesting paper which asks an important question about the preferences of publics in five NATO countries regarding support for Ukraine. The experiment is set up and analysed competently and the findings will be significant for a wide readership. The paper is hence a good fit for the journal. I would suggest four revisions, though: 1) how the paper is framed, 2) how it engages the study it replicates with a view to better framing the paper, 3) how to make the paper's theoretical assumptions and potential contributions clearer and 4) how to achieve a clearer exposition of the vignette experiment.

1) Framing

I would sharpen the framing of the paper to better draw out its political relevance. There is a bit of a tension between the claim that this study is critical because democracies have an "inherent vulnerability in deterrence games, i.e., the transparency of heterogeneous preferences undermining resolve" and the claim that "there exist no large-scale cross-country studies that scrutinize mass public attitudes among the Ukraine-supporting coalition for its resilience and resolve." Are these states currently really hindered in their quest to demonstrate resolve to Russia because they are democracies if the preference make-up of their populations is unknown? More importantly, the findings of the article are not consistently interpreted to derive clear implications for the ability of these countries to demonstrate resolve. The article is motivated with the importance of showing how democracies can "overcome" their vulnerabilities in deterrence games, which the article says little about. If Russia pays attention to mass attitudes in these countries when assessing NATO's resolve, then these countries can conceivably overcome the vulnerability of being democracies only in two ways, a) they can attempt to hide public discontent, or b) they can attempt to educate, cue and rally their publics. To do the latter, it is of course necessary to understand the structure of mass attitudes in the first place, so my point is not that there is no connection between the findings, NATO resolve, and the prospect of how this war will unfold, but the authors need to do a bit more to show this connection. I also wonder whether "demonstrating resolve" should really be the only/dominant motivation and frame of the paper (see next point).

2) Comparison with the Ukraine study

Maybe the framing of the paper could benefit from a more direct conversation with the study the authors replicate. This study (Dill, Howlett, Müller-Crepon AJPS 2023) asked Ukrainians what strategies for fighting Russia they support considering different costs and benefits. There is a democratic legitimacy/moral motivation for understanding how Ukrainians answer this question. Is there a similar (though maybe weaker) reason to understand the attitudes of countries that support Ukraine which could help motivate the paper? The authors argue that "A large faction of citizens unwilling to accept these human costs will consequently reduce resolve." It may be worth noting that, so far, the human costs of this war have accrued to Ukraine and Russia not to any of the surveyed countries and that the surveyed populations pay more attention to human costs that befall Ukrainians than material costs accruing to themselves. Is the implications of cost-sensitivity for resolve in NATO countries really the right frame for the paper?

In the interpretation of the results, what do we make of the differences in how NATO publics and Ukrainians assess different strategies for fighting Russia? NATO publics are more sensitive to Ukrainian casualties than Ukrainians, who the original study shows are very cost insensitive in their preference against concessions. What do we make of the facts that NATO countries pay more attention to human costs that do not directly befall their own countries a) than to material costs that do befall them, and as mentioned, b) pay more attention to Ukrainians' costs than Ukrainians? All the authors say about this is that this finding "hint[s] at potential conflict between Ukrainian and Western preferences over strategy." Of course, there is a prudential reason for Ukrainians being less sensitive to casualties, which is that they will bear the costs of concessions with Russia, also in terms of casualties. Moreover, Ukrainians also bear the costs to which they are insensitive as per the original study. That means they have more of a moral justification for disregarding loss of life among their own than NATO countries would have. In short, we should not have expected that attitudes in NATO countries follow the same logic and substantively track Ukrainians' attitudes, but that does not mean that these differences imply conflict. Much more could be said about the concrete theoretical and political relevance of the differences this study finds.

3) Theoretical assumptions and contributions

The paper has a lot of empirical material that it needs to discuss. Maybe as a result, the theory behind the experiment is arguably underdeveloped. The authors claim "we propose a new micro-foundation of autocratic containment and resolve of democratic citizens". What is that new micro-foundation? Are we confident that it is specific to containing autocratic regimes and rather than speaking generally to the structure of war support in Western countries? There are other studies that have found that in a direct comparison of human and material costs the former have a larger effect on war support (Krebs et al; Dill and Schubiger 2019). In fact, comparing these findings with the vast literature on Western war support what stands out as a new finding is that NATO publics pay almost as much attention to Ukrainian combatant deaths than to civilian casualties. However, since the implications of each strategy for Russia's further advancement, for instance into Poland or the Baltic states, are not spelled out in the experiment, the experiment does not seem to me to obviously be a test of people's views on the importance of containment of Russia or autocracies generally.

In the context of the paper's theoretical contribution, I would urge the authors to say more about the sub-group analysis, which I found very interesting, but which raised some questions for me. Are "pro-Western" and "anti-Western" attitudes distinct from the outcome measured or do they basically measure the same as the outcome (support for Ukraine to stay an independent Western oriented state against Russia)? Are these attitudinal measures that are stable and predictive in the context of other political outcomes? Does their predictive power imply that preferences here are not specifically about

Ukraine but an extension of people's general political world view? What does it mean to have "anti-Western attitudes" while living in the West? What were the questions used to establish these respondent traits, how were they aggregated or scored and what theory undergirds them? The paper goes into this on page 10, but as an aside after introducing the main result. Ideally the section would start with a definition and theoretical justification of how respondents were sorted into categories, how stable these categories are, their relevance in other contexts and what questions were used to group respondents.

Another under-explored avenue for a theoretical contribution might be the issue that publics across countries with divergent elite rhetoric and, to some extent policies, about this war have quite similar preferences at least on average. The paper touches on this but does not develop the issue. Maybe it would be too speculative unless paired with a systematic content analysis of elite rhetoric or media reports on Ukraine. So, the authors should feel free to ignore this point.

4) Exposition

Exposition-wise I had real trouble following the paper from page 11 onwards. The vignette experiment needs to be better explained. Was it the same respondent pool that took it, after the first experiment? What exactly do the vignettes say, is there an underlying theory for testing these different packages or is it just what the authors thought were likely choices? Does the fact that respondents seem to think more and longer-range capabilities increase Ukraine's chance of winning, winning more quickly, but also escalation risks really provide insight into the 'mechanism' behind their support for strategies (Figure 6)? The finding that pro- and anti-Western respondents view the consequences of these choices differently is interesting, but it is hardly indicative of why they are more/less supportive of helping Ukraine. Motivated reasoning would rather suggest that these expected consequences rationalise respondents' preferences. I am not saying the experiment does not reveal interesting findings, just that much more needs to be said to explain the experimental design and interpret the findings.

Reviewer #3

(Remarks to the Author)

The study addresses a highly relevant issue by examining how public opinion in Western countries is formed and evolves with respect to support for Ukraine. This is a crucial contribution to understanding strategic decisions by Western allies in response to Russian aggression. The topic is highly relevant in the current geopolitical context, particularly for decision-making within NATO and the EU.

A strong asset of this study is a fairly comprehensive survey conducted by the authors. The study employs a high-quality sampling strategy using YouGov's online panels in the United States, United Kingdom, Germany, France, and Italy. The sampling strategy and data collection methods appear to be robust and reflect the current distribution of political preferences in each country. The conjoint and vignette experimental designs are well structured and appropriate for addressing the research questions.

The results are presented clearly, with logical explanations of respondents' preferences and their variations. The use of conjoint experiments to quantitatively analyze the impact of different attributes on respondents' preferences is particularly noteworthy. The results effectively answer the research questions and demonstrate the significant role of public opinion in shaping Western support for Ukraine.

Er. The study conducts a good level of literature review, although there is not much literature available at the moment due to the ongoing situation. I think the authors adequately situate their findings within the existing body of research. By analyzing the relationship between public resolve and policy decisions from a new micro-foundational perspective, the study makes a certain level of theoretical contribution. This provides a good foundation for future research in this area.

The paper is generally well written and logically organized, with clear connections between sections. The flow of the research is smooth and the arguments are effectively communicated. However, some sections could benefit from additional elaboration for clarity.

The rationale for the selection of attribute levels and their operationalization could be more explicitly detailed. Providing more context on how these levels were determined would enhance the reader's understanding. In addition, the study could be strengthened by comparing its findings with those of other relevant research, such as Eurobarometer surveys, which have conducted extensive surveys on the Russia-Ukraine conflict with more accurate and comprehensive demographic profiles. There is not much academic research on this topic because the war is still ongoing. However, there are tons of reports and news articles on public opinion and attitudes towards the Russian-Ukrainian war. The authors will be able to find additional insights from these views.

The authors claim that this is the first large-scale cross-national study of public attitudes toward support for Ukraine. However, similar surveys have been conducted by think tanks such as the Chicago Council on Global Affairs and Pew Research. Acknowledging and differentiating from these existing studies would provide a clearer positioning of the current research.

The authors mentioned that the survey included respondents' characteristics (age, level of education, political orientation, etc.), but such details are completely absent from the analysis presented in the paper. It is well known that individual perceptions and views on the Russia-Ukraine war are significantly influenced by respondents' characteristics. The high

inflation caused by the Russia-Ukraine war, which has led to a decline in the purchasing power of European citizens and a general economic downturn, is well known. In addition, the overall rise in prices, including energy and food costs, has significantly altered the political landscape in Europe, contributing to increased support for far-right parties. During economic recessions caused by supply shocks, low-income groups tend to suffer more. Therefore, perceptions of the Russia-Ukraine war and willingness to support Ukraine are strongly influenced by individuals' economic situation. This study does not address these issues in the main text.

In addition, the survey should take into account whether respondents are natives or immigrants, which is also not mentioned in the study. Presenting survey results without distinguishing between economic and social groups can easily lead to biased results. Could these factors be the reason why support for Ukraine is not unconditional?

Overall, this study provides an important analysis of public opinion in Western countries regarding support for Ukraine. The methodology is sound and the results are clearly presented. However, the study would benefit from a more detailed explanation of the data, comparisons with other research, acknowledgement of existing studies in the field, and consideration of the economic and social groups of respondents. By addressing these issues, the study could provide more robust and comprehensive insights into the dynamics of public opinion and policy implications.

Version 1:

Reviewer comments:

Reviewer #1

(Remarks to the Author)

I write with regard to your question regarding the appeal for the rejected manuscript, "Citizen's Resolve Against Autocratic Aggression: Survey Experimental Evidence from Five NATO Countries on Supporting Ukraine." I read over the response memo while I do think that they were responsive to my concerns and suggestions, the revisions still leave my overall assessment of the paper unchanged. I remain somewhat unenthusiased about the suitability of the placement of this paper in a top general cross-field journal, like Nature Communications. It's a fine empirical paper with original data, but in the scope of things, I think it is much more appropriate for a subfield-specific journal in political science.

Reviewer #2

(Remarks to the Author)

The authors have done a commendable job in making the framing of the paper more compelling, explaining the vignette experiment better and anchoring the paper better in the existing literature. I am still not sure that calling what they study "a micro-foundation of resolve" is completely accurate or necessarily the best way of explaining the theoretical angle of the paper, but I am convinced that the paper has an important contribution to make to the study of resolve in IR and the study of mass attitudes on war in comp gov. I also think the paper could hardly be more timely. Of course, given that the appeal of the paper is bound up with its current political relevance, it will be critical for the authors to add a brief reflection on recent developments. In a sense the likely/imminent withdrawal of US aid from Ukraine seems to be a phenomenon largely disconnected from public opinion dynamics in the United States. And yet, the reasoning of the US administration has a decidedly populist flavor, ostensibly appealing to the need to "stand up for the American people". So in short, I do not think this detracts from the value of the paper, which I would be enthusiastic to see in print, but it should briefly be addressed. Along similar lines, a replication of the AJPS study of Ukrainian attitudes that the authors rely on now suggests that their attitudes remain steadfast along some lines, but have changed/weakened along others. It might be worth gesturing to this as well.

Reviewer #3

(Remarks to the Author)

I have reviewed the authors' response to my comments and their revised manuscript, and think they have made a sincere effort to address the key points while maintaining the overall focus and structure of their study. They have clarified their methodological choices, particularly in explaining the choice of attributes and their operationalization, and they have also acknowledged relevant prior research, including comparisons with existing studies. In addition, they have improved the clarity of their argument by refining sections such as the introduction and methodology, making the manuscript more coherent and structured.

While some areas, such as subgroup analyses based on economic or social background, could be further explored in future research, I find that the authors have provided reasonable justifications for their current approach. Their revisions adequately strengthen the manuscript without altering its core contributions. Given these improvements, I think the paper is now ready for publication.

Response Memo

Citizen's Resolve Against Autocratic Aggression: Survey Experimental Evidence from Five NATO Countries on Supporting Ukraine

January 8, 2025

Dear Referees,

We thank you for the work entailed in reassessing the revised version of our manuscript NCOMMS-24-39785-T “Citizen’s Resolve Against Autocratic Aggression: Survey Experimental Evidence from Five NATO Countries on Supporting Ukraine”. We are grateful for your very helpful feedback on the previous version of our manuscript. In light of your suggestions, we have carefully and extensively revised the paper and – given the high topicality – remain firmly committed to implementing whatever additional changes might be necessary to make the paper acceptable for publication in Nature Communications.

In what follows, we reproduce the comments as given in order (in standard font, just adding a cross-numbering) to respond to them directly (in **bold** font) and discuss the respective changes made in the manuscript. For ease of refereeing, we incorporate core revisions of the manuscript directly in this memo (in *italics*).¹ Also, we attach a version delineating track changes compared to the original submission.² You find our replies to the comments of Reviewer #1 from page 2, of Reviewer #2 from page 8, and of Reviewer #3 from page 18.

We very much look forward to hearing from you again.

Yours sincerely,

The authors

¹Please note that in these *italic* passages, the content of reference numbers is identical to the manuscript, but numbering differs.

²Please note that track changes in LaTeX do not differentiate between content that is shifted within the manuscript, and content that is deleted (this mostly concerns some colored paragraphs in the introduction and in the new argument section). Also, all figures are marked as blue (while they actually are unchanged), which we have no influence over. Only one, additional SI figure, is new, otherwise all empirical material remained the same.

Comments by Reviewer #1

R1-1 This paper employs a five-country sample to explore the impact of hypothetical changes in the military situation in Ukraine on expressed support for the Ukrainian effort. The authors employ a large-scale conjoint experiment to assess perceived support across a variety of human and economic costs, as well as ultimate strategic aims. They also use a vignette experiment to try to assess the mechanisms by which respondents assess the perceived consequences of action. The authors find that variation in the human costs of war on the Ukrainian side lead to different levels of support, as do various war aims (containment of aggression and sovereignty). Economic costs, however, are largely unrelated. The pattern of results is very similar across countries, indicating that while some differences may exist in the structure of the preferences of citizens of different nations, there is much more similarity across the U.S. and Western Europe.

We very much thank R1 for her/his detailed assessment of our manuscript. We carefully considered his/her comments as detailed below and reworked the manuscript subsequent to the suggestions provided.

R1-2 This paper is first and foremost an empirical paper. The underlying mechanisms are drawn from other work and there is little new in terms of theoretic advance. Whether such a paper is worthy of publication in Nature Communication is a decision that the editor must make. Personally, I think that the advancement of knowledge through empirical work is a worth endeavor and my comments should be read as suggestions to improve the description and presentation of the results.

We thank R1 for this assessment. We agree that the major objective of our manuscript is to make an important empirical contribution to a highly topical case. But at the same time we address for the first time the general and essential theoretical question of democratic resolve to autocratic military challenges to the current liberal world order. This is exemplified with the Ukrainian case. In the introduction, we now made this clear by explicitly including that we are the first to investigate, with survey-experimental cross-country experiments citizens' attitudes regarding Ukraine strategies by their own governments, the extent of cross-country heterogeneity or similarity, and the reasons behind this. Also, we link there to the new SI Section 1, which summarizes recent empirical literature from mostly single-country public opinion surveys regarding the Ukraine war – we improve over this literature both regarding internal and external validity with our research design. While our research design (for the conjoint part) is strongly inspired by Dill et al. (2024), the fact that Western countries' populations assess a foreign invasion on foreign soil changes theoretical priors regarding preference formation – which we provide. To be more concrete, we would like to emphasize that we see our manuscript making actually relevant theoretical contributions by: 1) linking the war sup-

port literature with its nuanced findings on citizens' preferences to the literature on deterrence games (usually taking a US-related perspective); 2) extending the war support literature to a case of support for a war on foreign soil without direct military involvement, with new findings how strategic risks, economic costs and human life is weighted and traded-off by citizens in such a case; 3) by considering the extent to which citizens support the current liberal world order as important moderator for resolve against autocratic challenges, and testing this empirically. This now reflects in more detail in the reworked framing of the introduction, the new argument section, the new introduction to sections 3.1.3 and 3.2, and the reworked interpretation of our results in the conclusion. Given the extensive changes, we would like to refer the reviewer to these sections for details and refrain from copy-pasting these changes also here.

R1-3 I think that this paper is interesting and tackles an important (and current) topic. I also appreciate the scope of the study – the authors ran five well-powered surveys to compare preferences across countries and the data collection itself is an impressive undertaking. However, I think that the authors make a stronger causal claim than is warranted by the design of the study. I suggest that the authors tone back their language in a few places and discuss the limitations of employing hypothetical scenarios in a real-world (and ongoing) conflict.

We thank R1 for including this note of caution. We agree that ‘hypotheticality’ is a relevant concern in such survey experiments. Note, however, that the (only, to our knowledge) proper study comparing hypothetical conjoint choices to natural experimental actual behavior indicates high comparability [1] (see point below for a broader discussion). Still, we agree that given this rather slim evidence body, the extent to which results would travel to real-world (e.g., referendum or voting) choices must be carefully assessed. Subsequently, we indicate this explicitly now 1) in the manuscript (conclusion), and 2) in more detail within a new extensive Methods section 4 on strengths and limitations. This section now discusses the strengths and limitations of conjoint survey experiments regarding the simultaneous comparability over multiple attributes, social desirability, bias from ‘masking’, hypotheticality, and spillovers. Its summary in the conclusion reads as follows [parts written in *italics* in the manuscript are underlined here]:

We conclude by outlining the strengths and limitations of our survey-experimental approach (see Methods, Section 4 for details). The design allows us to assess, with a causal interpretation, the relative importance of the theoretically derived attributes and as weighted by respondents at the time of our survey. Given our monitoring of public debates at that time, we are confident we inquired about aspects 1) most salient to public debate, 2) that can realistically be assessed by respondents. Indeed, an experiment is

always a stylized depiction of reality,³, i.e., the potential of masking of preferences on mentioned by unmentioned attributes has to be taken into account in interpretation [2]. A relevant second concern is the hypotheticality of decision-making in such experiments, given drastic real-world changes of attributes are usually more salient, and take place in a complex changing broader information environment. At the same time, past research indicates survey-experimental decision-making can mirror real-world behavior remarkably well [3], and that swings in public opinion based on conjoint experimental conditions can represent swings based on real-world changes [see 4, studying weapon transfers, relating to Ukraine]. Last, factorial survey experiments as our conjoint show a very desirable property in likely minimizing social desirability bias [5]. Overall, our focus on citizen preferences is in line with recent scholarship that indicates that foreign policies attract citizens' attention in a similar way as domestic policy, i.e., that citizens form preferences consistently in both domains [6–8], and that citizens' opinions, as measured in surveys, seemingly matter for policy debates, political mandates [9–12], and ultimately for policy processes [13, 14] and elite decision making [15–19].

R1-4 On page 5, the authors claim that their results are strongly identified because the draw on “conjoint and vignette experiments that allow the identification of causal effects.” This is very strong language. I would note that the entire experimental setup relies on individuals responding to hypothetical scenarios, not changes in real-world events. Some political scientists and economists would argue that such a design can only measure “cheap talk” hypothetical preferences. I am not one of them, but I do think the authors need to recognize and acknowledge the potential limitations of employing hypothetical scenarios. This piece by Susan Hyde in the Annual Review of Political Science (<https://doi.org/10.1146/annurev-polisci-020614-094854>) is somewhat old but could be a good place to start a discussion of the strengths and limitations of survey experiments.

We very much thank the author for pointing us to Hyde [20], which we took up, along with a longer discussion (see point above) on strengths and limitations of survey experiments. For example, we now note in the new Methods section 4 on strenghts and limitations that *The survey-experimental approach we apply allows us to relate experimenter-induced variation in question-wording to differences in answering behavior in a causal way [3, 21]. This improves over a number of known concerns from traditional surveys eliciting population preferences [22], and is useful insofar as we explicitly aim at testing micro-foundations of International Politics theories with a public opinion survey – with a sample explicitly targeted at being population representative also beyond standard quotas, and going beyond the most regularly studied US population [20]. At the same time, our approach has relevant limitations. [...]*

Among the discussion of limitations (see Methods section 4), we also included a specific section on hypotheticality, which now reads: *Fourth, we fielded sur-*

³We did not, for example, spell out civilian suffering in the occupied Russian territories; neither did we spell out the risk of a Russian invasion of NATO territory

vey questions on hypothetical scenarios. With such scenarios, there might be concerns that hypotheticality induces bias in responses in such choice experiments [23]. However, Hainmueller et al. [24] show that survey-experimental results can mimic consequential real-world choices remarkably well, and best in the paired profile choice design that we apply.⁴ This strongly indicates that survey experiments are actually more than just ‘cheap talk’. We are not aware of much additional literature explicitly providing for tests in this regard, all the less regarding topics of international politics and war. Noteworthy is Rudolph et al. [4, p. 723], given its direct relation to our case: They are able to compare estimates from conjoint survey experiments on general preferences regarding stylized arms exports decisions fielded in a pre-Russian-Ukraine war time, but depicting scenarios akin to this war as well as without aggression. Comparing changes in marginal means to the observed swing in public opinion regarding whether arms should be provided to Ukraine or not as measured through public opinion surveys just before and after war onset yields high comparability. Overall, this is comforting, as it indicates that survey experiments have ‘ecological validity,’ i.e., respondents’ hypothetical choices in a low-consequence environment travel to their real-world preference formation and decision-making.

R1-5 On a related note, I found the vignette experiment interesting, but I was not convinced that the various dependent variables were strong tests of the mechanism at work. The perceived consequences of action (chance of Ukraine winning, quicker war ending) might be causally prior to overall support. But it could well be that the consequences are themselves endogenous to support levels. The exchange between Berinsky and Druckman on the one hand, and Gelpi and Reifler on the other concerning the causal impact of war success (in *Public Opinion Quarterly* – starting with, Vol. 71, No. 1, Spring 2007, pp. 126–141) could be helpful in reframing the strong causal claims made in the paper.

We are grateful to R1 for pointing this out. We consequently reframed the vignette section on what we have causal evidence for: the comparison in stated agreement with different types of military and economic aid, and the comparison in perceived consequences between these types of aid. This comparison has a causal interpretation (regarding stated preferences) as our design prevents spillovers between respondents’ assessments of one type of aid compared to the other. At the same time, we note that we cannot causally identify *why* respondents support one type of aid over the other, exactly for the reason outlined by R1. We hence also no longer speak of ‘mechanisms’. This is now prominently introduced at the beginning of the section as follows:

Our aim is to causally compare stated support levels and stated perceived consequences of aid by the respondents’ government to Ukraine. Given respondents are asked about one vignette only, our design prevents spillovers in respondents’ assessments of one type of

⁴In their case regarding citizens’ attitudes toward immigration, i.e., a topic highly prone to social desirability bias.

aid to another.⁵

It is also summarized, with reference to the debate the reviewer pointed us to, at the end of section 3.2.2. This reads as:

While insightful, whether these perceptions of consequences are causally prior to stated agreement to aid is unclear. We cannot disentangle with our design whether respondents agree with certain types of aid because of perceived consequences or whether they adapt perceptions of consequences to what they agree with. Berinsky [25] convincingly argue that aspects like expected military death, the existence of a rightful cause, or perceived success probabilities may causally determine support for military action, it is just as well probably that underlying respondent characteristics (like partisan predispositions) jointly affect perceived consequences of aid types and agreement with aid itself. To establish causal direction, we would call for additional experiments [see also 26].

The remainder of this section is reframed, now more carefully speaking of stated support, and more strongly focusing on the comparison between vignette conditions.

R1-6 I also had a few specific notes: 1. Section 2.2 needs to be expanded. In particular, the discussion of the theoretic and empirical setup of the vignette experiment should be fleshed out. The links to the various theories discussed earlier in the paper were not clear and I had to read the section several times to understand exactly what the authors were doing (and even then, I was sometimes confused).

We thank R1 for pointing this out. We revised section 2.2 (which is now numbered 3.2) considerably, strengthening both the theoretical argument what the vignette experiment actually tests, the explanation of the design, and the interpretation of its results.

Given the extensive revisions, we point the reviewer to this revised section for details.

R1-7 2. In the appendix, the authors make note of recent criticisms of the use of ACMEs to assess majority preferences and subgroup preferences. They find the pattern of results hold when using ACPs and AFCEPs. I would mention this fact in the body of the paper to avoid confusion, given the salience of recent debates about this matter.

We thank R1 for highlighting this aspect. This is now taken up in new Footnote 21, which reads as follows:

Methods section 5 provides for estimates from Average Component Effects (ACPs, Ganter [27] and Average Feature Choice Probabilities (AFCEPs, Abramson et al. [28] given recent debates on AMCE interpretation. These show the same pattern of results as our AMCE/marginal means plots as presented below.

⁵Note that we can not causally identify *why* respondents agree with certain types of aid. Perceived consequences are likely causally prior to agreement to aid, but could as well, e.g., via motivated reasoning, be endogenous to support.

Overall, we again want to thank R1 for the detailed reading of our manuscript and the insightful comments we received. Incorporating these comments strengthened our manuscript considerably.

Comments by Reviewer #2

R2-1 This is an interesting paper which asks an important question about the preferences of publics in five NATO countries regarding support for Ukraine. The experiment is set up and analysed competently and the findings will be significant for a wide readership. The paper is hence a good fit for the journal. I would suggest four revisions, though: 1) how the paper is framed, 2) how it engages the study it replicates with a view to better framing the paper, 3) how to make the paper's theoretical assumptions and potential contributions clearer and 4) how to achieve a clearer exposition of the vignette experiment.

We thank R2 for her/his detailed reading and the positive assessment of our manuscript. We have carefully considered and incorporated the helpful comments we received, which have relevantly strengthened our manuscript.

R2-2 1) Framing I would sharpen the framing of the paper to better draw out its political relevance. There is a bit of a tension between the claim that this study is critical because democracies have an “inherent vulnerability in deterrence games, i.e., the transparency of heterogeneous preferences undermining resolve” and the claim that “there exist no large-scale cross-country studies that scrutinize mass public attitudes among the Ukraine-supporting coalition for its resilience and resolve.” Are these states currently really hindered in their quest to demonstrate resolve to Russia because they are democracies if the preference make-up of their populations is unknown?

We thank R2 for this remark. We agree that the framing of the paper can be sharpened (and at the same time theoretically broadened). We have done so as outlined in our detailed response to the comments below. As a side note, the tension R2 perceives between the two sentences quoted above is no actual tension, in our opinion. If we understand R2 correctly, he/she infers that our claim democracies are constrained by their heterogeneity is incorrect, given we claim this was unknown before our study. We argue that the scientific public and the general public may not have had access to a study on this matter with high internal and external validity (as we now provide). But Western governments and also autocratic countries' governments should via intelligence, private campaign preparation, and classified government population monitoring already have knowledge of the findings we present. We have sharpened our wording to preclude readers perceive this tension by rephrasing to *[no ...] scientific studies*, and referencing our contribution to ... *thereby informing both scholarly and public debate on page 3*.

R2-3 More importantly, the findings of the article are not consistently interpreted to derive clear implications for the ability of these countries to demonstrate resolve. The article is motivated with the importance of showing how democracies can “overcome” their vulnerabilities in deterrence games, which the article says little about. If Russia pays attention to mass attitudes in these countries when assessing NATO's resolve, then these countries

can conceivably overcome the vulnerability of being democracies only in two ways, a) they can attempt to hide public discontent, or b) they can attempt to educate, cue and rally their publics. To do the latter, it is of course necessary to understand the structure of mass attitudes in the first place, so my point is not that there is no connection between the findings, NATO resolve, and the prospect of how this war will unfold, but the authors need to do a bit more to show this connection.

We thank R2 for these thoughtful reflections. In response, we have first of all rephrased the corresponding sentence in the introduction to...the important question of how extensive the inherent vulnerability of democracies in deterrence games is, i.e., how prevalent preferences undermining resolve are.

Second, and more broadly, we extensively reworked our discussion of practical implications in the conclusion, also taking up the helpful suggestion of R2. This now reads as follows:

Regarding the practical debate, when Western countries decide on support strategies, they have to take into consideration that popular support is not unconditional, i.e., that (popular) resolve has boundaries. Given foreign intelligence is likely both informed about these boundaries, and might even, in the age of social media, try to influence them via stealth misinformation campaigns [29], this emphasizes the importance of communication between governments and their population on these matters. The credibility and content of elite cues [30], how united elites portray attempt to educate and rally their publics, then becomes pivotal. Regarding current debates, our study indicates that the boundaries that citizens draw on Ukraine support seem to relevantly differ from their governments, constraining the actions of the latter. For example, minimizing Ukrainian casualties should be a top priority for Western governments if they want to sustain a strong resolve. Preventing nuclear escalation at the same time seems of utmost importance. Here, counter-communication against autocratic signaling that strategically manipulates perceived risks of nuclear escalation seems pivotal for sustained support.⁶ Given economic costs of aid, particularly of military aid, seem not to be a relevant constraint, Western governments should not be constrained to ramp up military aid considerably. This also indicates that party-level contestation, e.g., in the US surrounding domestic economic opportunity costs, does not resonate similarly in public sentiment. Future research could probe here whether different depictions of economic costs (e.g., on a household basis) yield similar results.

R2-4 I also wonder whether “demonstrating resolve” should really be the only/dominant motivation and frame of the paper (see next point). 2) Comparison with the Ukraine study Maybe the framing of the paper could benefit from a more direct conversation with the study the authors replicate. This study (Dill, Howlett, Müller-Crepon AJPS 2023) asked Ukrainians what strategies for fighting Russia they support considering different costs and benefits. There is a democratic legitimacy/moral motivation for understanding how

⁶See <https://www.csis.org/analysis/russian-nuclear-calibration-war-ukraine>.

Ukrainians answer this question. Is there a similar (though maybe weaker) reason to understand the attitudes of countries that support Ukraine which could help motivate the paper? The authors argue that “A large fraction of citizens unwilling to accept these human costs will consequently reduce resolve.” It may be worth noting that, so far, the human costs of this war have accrued to Ukraine and Russia not to any of the surveyed countries and that the surveyed populations pay more attention to human costs that befall Ukrainians than material costs accruing to themselves. Is the implications of cost-sensitivity for resolve in NATO countries really the right frame for the paper?

We thank R2 for this interesting consideration. While we kept our general frame, we agree that it need not be the only frame to consider. We subsequently emphasized the relevance of our question regarding the broader question how citizens form preferences akin to war support and regarding practical strategies in Ukraine, as the revised introduction shows:

The literature on mass popular support for war provides important leads. This literature shows that economic costs, moral constraints, and strategic risks are relevant factors for supporting a country’s military operations [31, 32], varying by context conditions. This allows to shed light on citizens’ preference formation regarding foreign policy more generally, and to derive specific conditions under which public opinion can undermine resolve for domestic wars. But this framework has, to our knowledge, never been applied to decisions of external aid by not directly involved third countries for wars fought on foreign soil. The case of Ukraine is particularly prominent in this regard, as economic costs and strategic risks of escalation are relevant domestic concerns, while the human costs of this war and the immediate consequences of any agreement are only born by the population of the foreign country.

Note, however, that we consider our proposed framing of ‘resolve’ a so far neglected aspect in the scientific discussion. It has been very important in the fight against Nazism as we demonstrate by referring to early survey evidence from Berinsky [33]. Notably, this can only be investigated by assessing public opinion of the countries that actually have the military capabilities to take this fight, i.e., the US and the most powerful European countries as we do. The approach by Dill *et al.* [34] certainly makes very important contributions, and directly inspired us to this study, but we would at the same time like to relate to and use our data for a deeper theoretical foundation on the conditions under which the countries backing the current liberal order can challenge external interveners.

R2-5 In the interpretation of the results, what do we make of the differences in how NATO publics and Ukrainians assess different strategies for fighting Russia? NATO publics are more sensitive to Ukrainian casualties than Ukrainians, who the original study shows are very cost insensitive in their preference against concessions. What do we make of the facts

that NATO countries pay more attention to human costs that do not directly befall their own countries a) than to material costs that do befall them, and as mentioned, b) pay more attention to Ukrainians' costs than Ukrainians? All the authors say about this is that this finding "hint[s] at potential conflict between Ukrainian and Western preferences over strategy." Of course, there is a prudential reason for Ukrainians being less sensitive to casualties, which is that they will bear the costs of concessions with Russia, also in terms of casualties. Moreover, Ukrainians also bear the costs to which they are insensitive as per the original study. That means they have more of a moral justification for disregarding loss of life among their own than NATO countries would have. In short, we should not have expected that attitudes in NATO countries follow the same logic and substantively track Ukrainians' attitudes, but that does not mean that these differences imply conflict. Much more could be said about the concrete theoretical and political relevance of the differences this study finds.

We thank R2 for pointing this out and allowing us to sharpen the interpretation of our findings. We took this up both in the introduction, where we now emphasize that *Second, any resolve Western countries or their populations are willing to exhibit is indeed bounded by preferences within Ukraine – military aid can only be effective as long as this population wants to sustain their fight. At the same time, Ukrainian domestic resolve is bounded by the support it receives from Western countries, as its military capabilities otherwise cannot keep up with those of the aggressor given the current war of attrition [35]. Related research by Dill et al. [34] shows that the attacked Ukrainian population – in line with its government – is willing to fight ‘at any cost’ against the Russian aggression, accepting large strategic risks (i.e., nuclear escalation) and human costs (among military and civilians) for upholding their full sovereignty and territorial integrity. But to what extent do the attitudes of Western populations align with this view? Descriptive evidence from surveys with European citizens indicates that Western populations show differentiated preferences [36].⁷ But to our knowledge, there exist no large-scale cross-country scientific studies that scrutinize mass public attitudes among the Ukraine-supporting coalition for its resilience and resolve, thereby informing both scholarly and public debate.*

Also, we reconsider this in the conclusion, emphasizing that *Last, it is insightful to compare our findings to the study by Dill et al. [34]⁸, regarding preferences in the Ukrainian population. After all, resolve against the Russian aggression has two necessary conditions: Western support and Ukrainian willingness to uphold the fight. Our findings contrast to an almost unconditional support in favor of defending territory and sovereignty against Russia among the Ukrainian population (in spite of human costs and escalation risk). That the populations of the Western support coalition seemingly give larger weight to human costs abroad compared to the affected population seems surprising – one potential*

⁷See also Ash *et al.* [37].

⁸Whose design is directly comparable as we fielded several attributes they inquired.

explanation could be that it is only direct affectedness that can extend moral legitimacy to take this very hard trade-off. But at the same time, the Ukrainian population is also more strongly aware of the human costs of giving in to Russian aggression, something external to the experiments. Regarding potential agreements for a freezing of the conflict or even a lasting peace, the Ukrainian population rejects territorial concessions of only the Crimean peninsula nearly as strongly as concessions as of the current line of conflict (relative to full integrity) – among Western populations, this differs by more than a factor of 2. This indicates a potential conflict between deals that could be acceptable to Western populations compared to the Ukrainian citizenry.

R2-6 3) Theoretical assumptions and contributions The paper has a lot of empirical material that it needs to discuss. Maybe as a result, the theory behind the experiment is arguably underdeveloped. The authors claim “we propose a new micro-foundation of autocratic containment and resolve of democratic citizens”. What is that new micro-foundation? Are we confident that it is specific to containing autocratic regimes and rather than speaking generally to the structure of war support in Western countries? There are other studies that have found that in a direct comparison of human and material costs the former have a larger effect on war support (Krebs et al; Dill and Schubiger 2019). In fact, comparing these findings with the vast literature on Western war support what stands out as a new finding is that NATO publics pay almost as much attention to Ukrainian combatant deaths than to civilian casualties. However, since the implications of each strategy for Russia’s further advancement, for instance into Poland or the Baltic states, are not spelled out in the experiment, the experiment does not seem to me to obviously be a test of people’s views on the importance of containment of Russia or autocracies generally.

We thank R2 for highlighting this relevant literature and the indicated limitations of generalizability. In consequence, we have intertwined our angle of resolve more deeply with the war support literature.

This concerns on the one hand the new section 2 (argument), where we now emphasize that our discussion regarding normative, economic and strategic aspects of preference formation (for a foreign war on foreign soil) also relevantly link to the war support literature (whose literature we of course strongly build upon. We refer the reviewer to the changes made in this new section (which early was partly contained in the former introduction).

This now also reflects in the conclusion, where we emphasize the contributions we make to this literature more directly: extending to a case where the war is fought on foreign soil and the question is whether to support an invaded country; where human suffering is not involving killing of or by soldiers of the external intervenor; where the threat of nuclear escalation is relevant; while investigating preferences in a usually US focused literature that rarely

employs large-scale comparative designs.

These three findings shed light on recent theoretical and practical debates in two ways: Regarding the theoretical debate, our case supports arguments that citizens take moral considerations extensively into account when forming preferences [e.g., 6]. We extend this literature, however, assessing the case of a conflict on foreign soil invaded by foreign troops, where human suffering is spatially very distant (not involving killing of or by domestic soldiers). Still, in our case, human life abroad seems to be valued much higher by Western citizens compared to domestic economic costs. While other research emphasized materialistic preferences and economic calculi to be highly relevant for foreign policy preference formation [see e.g. 38–40], this does not reflect here. This is in line with recent arguments on the relevance of military and civilian casualties in the war support literature. For example, Kreps & Maxey [41] show that harm to civilians and security considerations, but not economic costs, are strong determinants of (US) citizens' attitudes towards humanitarian interventions; Dill & Schubiger [42] show that – in line with principles of international law – (US) citizens despise civilian casualties, but not enemy military casualties while caring strongest about domestic military casualties. Our research corroborates these findings within the context of a war fought on foreign soil, also indicating strong caring for Ukrainian combatant death at par with care for Ukrainian civilian life. Notably, we are the first to consider a confrontation with a potent nuclear power, where the strategic risk of nuclear escalation is plausible, and weighted similarly strongly by respondents. Future research could probe whether this is because of humanitarian concerns, or concerns of the war extending to the domestic realm. Last, our study crucially extends findings from the war support literature with its heavy US focus (for exceptions, see, e.g. Dill et al. [43]) to populations of four NATO middle powers.

Last, regarding generalizability to Eastern European countries we agree with R2 that this is a limitation. At the same time, we had to put limitations somewhere, and for us, it was important that our country selection represents the most capable countries. This is now remarked at the end of this paragraph in the conclusion.

While these NATO countries are practically highly relevant, given they exhibit the largest shares of military exports, thus being representative of military support capabilities, future research would need to assess to what extent our findings generalize to Eastern European countries sharing a border with Ukraine and/or Russia.

R2-7 In the context of the paper's theoretical contribution, I would urge the authors to say more about the sub-group analysis, which I found very interesting, but which raised some questions for me. Are “pro-Western” and “anti-Western” attitudes distinct from the outcome measured or do they basically measure the same as the outcome (support for Ukraine to stay an independent Western oriented state against Russia)? Are these attitudinal measures that are stable and predictive in the context of other political outcomes? Does

their predictive power imply that preferences here are not specifically about Ukraine but an extension of people’s general political world view? What does it mean to have “anti-Western attitudes” while living in the West? What were the questions used to establish these respondent traits, how were they aggregated or scored and what theory undergirds them? The paper goes into this on page 10, but as an aside after introducing the main result. Ideally the section would start with a definition and theoretical justification of how respondents were sorted into categories, how stable these categories are, their relevance in other contexts and what questions were used to group respondents.

We thank R2 for these additional questions and the opportunity to sharpen our argumentation. We now introduce this section (3.1.3) much more extensively both regarding our argumentation and regarding its empirical design. The definition and theoretical justification, as well as the questions used to group respondents, are explained in the following part of section 3.1.3:

However, given we inquire (Western democracies’) resolve against (Russian) autocratic aggression, we propose that the extent to which Western citizens themselves are rooted in the liberal, rule-based order promoted by Western democracies is likely at the core of within-country heterogeneity. This is in line with broader arguments that the contestation of the current liberal world order not only stems from actors outside the ‘West’, but also from within, i.e., from factions in Western societies, particularly far-left and far-right movements [44]. Notably, the backbone of the current liberal world order is the transatlantic security framework based on a US-led NATO alliance seeking to expand and defend liberal internationalism. To capture whether respondents align or not align with liberal internationalism, we therefore distinguish respondents by whether they support vs. renounce transatlanticism. These value dispositions are a much more general account of preferences regarding the role of the Western alliance compared to our concrete Ukraine strategy assessments. Concretely, we derive a composite index from a principal components analysis (PCA). The index is based on three questions regarding respondents’ “overall opinion of NATO”, their (dis)agreement that “American imperialism is the real threat to world peace” and their understanding that “Russia feels threatened by the West” (for details, see Methods). Extensive exploratory analyses indicate that this is the major divide between citizens’ positions related to Ukraine policies. For the subsequent presentation of results, we split respondents into quartiles and estimate MMs for these.

Furthermore, a more detailed description of the identified groups (which also provides indication of the stability of these categories) is given in the following part of this section:

Linking to the statements that made up the PCA score, the 25% of respondents with strong pro-Western attitudes overwhelmingly reject that they can understand Russia feels threatened (89%), reject that US imperialism is a danger to world peace (83%), and report a positive stance towards NATO (87%); while the 25% of respondents with strong anti-western attitudes overwhelmingly agree that Russia feels threatened (74%), agree on US

*imperialism (76%) and report a negative (50%) or torn (38%) stance towards NATO. SI Figure 6 presents evidence that both the composite index and its components show a u-curved relationship with the standard left-right axis: scores are higher for both the extreme right and the extreme left, pointing to a specific coalition of anti-democratic publics within the Western democracies, and to threats to resolve from both the radical left and right. SI Table 4 goes into more detail of who these anti-western respondents are. Here, we regress anti-western preferences (PCA index) on sociodemographics of respondents, as well as political attitudes, and values. From model 1, the association to socio-demographics, we find that anti-western attitudes are relevantly associated with country of origin (lower in the UK, higher in France and Italy, compared to Germany and the US), with age (lower in age cohorts above 45), with education (lower for high-educated respondents), higher with worry over the personal or country-wide state of the economy, associated with religious affiliation (lower with Protestant, higher with Orthodox or Muslim faith, compared to no stated religious affiliation or Catholic faith); there is no discernible association with gender or work status (*ceteris paribus*). Adding political attitudes to this model (model 2), the above-mentioned associations remain robust. Ideology again shows a u-shaped relationship (though significant only for the political right), while stated turnout for the last general election relates negatively to anti-Western attitudes; political interest shows no stable relation. Adding additional value configurations of respondents to these models, foreign policy values are associated relevantly: isolationist attitudes correlate positively, while militant interventionist, but also cooperative internationalist values correlate negatively with anti-western attitudes (model 3). Last, attitudes towards war and peace (with higher values indicating anti-war, or pro-peace attitudes) correlate negatively with anti-western attitudes (model 4). Compared to the standard deviation of the dependent variable, effect sizes are in-between 6% (French origin, significant on the 10% level, model 1) and 49% (Orthodox faith, model 2) of this standard deviation. Hence, they can reach a relevant magnitude. Taken together, these results indicate clearly, on the one hand, that idiosyncratic country-level factors beyond the set of correlates we used lead to variation in resolve against aggression between the five countries; on the other hand, within countries, the shown associations provide some context for which societal coalitions for pro- and anti-Ukraine support could form.*

R2-8 Another under-explored avenue for a theoretical contribution might be the issue that publics across countries with divergent elite rhetoric and, to some extent policies, about this war have quite similar preferences at least on average. The paper touches on this but does not develop the issue. Maybe it would be too speculative unless paired with a systematic content analysis of elite rhetoric or media reports on Ukraine. So, the authors should feel free to ignore this point.

We agree with R2 that this would be a worthwhile avenue, outside the scope of our already very dense manuscript, however. However, taking up the helpful lead of the reviewer, and to point readers to a relevant research question that could be answered with our replication data set we now included in this section

3.1.3:

The question of why this is the case would require an extensive linkage of our data with data on elite rhetoric and/or the media coverage of the Ukraine war between these five countries at the time of our survey. This question is important, but beyond the scope of this manuscript, which is why we leave it to future research.

R2-9 4) Exposition Exposition-wise I had real trouble following the paper from page 11 onwards. The vignette experiment needs to be better explained. Was it the same respondent pool that took it, after the first experiment? What exactly do the vignettes say, is there an underlying theory for testing these different packages or is it just what the authors thought were likely choices? Does the fact that respondents seem to think more and longer-range capabilities increase Ukraine’s chance of winning, winning more quickly, but also escalation risks really provide insight into the ‘mechanism’ behind their support for strategies (Figure 6)? The finding that pro- and anti-Western respondents view the consequences of these choices differently is interesting, but it is hardly indicative of why they are more/less supportive of helping Ukraine. Motivated reasoning would rather suggest that these expected consequences rationalise respondents’ preferences. I am not saying the experiment does not reveal interesting findings, just that much more needs to be said to explain the experimental design and interpret the findings.

We thank R2 for pointing this out. In response, we have thoroughly reworked this section (3.2.), introducing its aims, argumentation and design more clearly up-front, and also revisiting our interpretations – particularly of how agreement to aid and perceptions of consequences relate. These changes were mainly taken up in the revised introduction of the vignette section [parts written in *italics* in the manuscript are underlined here]:

We extend our results on abstract Ukraine strategies with original evidence regarding (dis)agreement of respondents to providing specific types of aid, as well as perceived consequences of such aid. This not only directly informs heated policy debates in the countries under study on how to assist Ukraine concretely but gives us an indication by which means and for what reasons respondents want to stand firm with Ukraine or not. We inquire the two core dimensions of aid, military and economic, and presented respondents the most prominently discussed aid options in public debate at the time of our survey.

We chose a 1x6 vignette design for this question. Our aim is to causally compare stated support levels and stated perceived consequences of aid by the respondents’ government to Ukraine. Given respondents are asked about one vignette only, our design prevents spillovers in respondents’ assessments of one type of aid to another.⁹ This experiment was fielded after the conjoint experiment. Figure 5 provides details.

⁹Note that we can not causally identify *why* respondents agree with certain types of aid. Perceived consequences are likely causally prior to agreement to aid, but could as well, e.g., via motivated reasoning, be endogenous to support.

Our theoretical justification for this vignette lies in the heavy contestation of this aid within Western democracies. Western governments provide this aid with the intention of enabling Ukraine to defend itself, i.e., to increase its winning chances and end the war. We propose that citizens will likely agree to aid if caring for outcomes in Ukraine. Given the results of the conjoint experiment, respondents with anti-Western attitudes should therefore disfavor, and respondents with pro-Western attitudes agree to aid – and all the more the more they perceive its consequences to be beneficial for Ukraine.

But note that the conjoint experiments indicates that respondents also relevantly take human costs and strategic risks into account. It is an empirical question, therefore, to what extent they at the same time perceive negative consequences regarding escalation risk and human suffering (and also domestic economic opportunity costs) and whether overall negative consequences on these dimensions can motivate disagreement. If negative consequences have large weight, this would clearly undermine Western governments' ability to supply robust aid to Ukraine, and thereby directly diminish their potential resolve, restricting them to certain aid types.

Note that we can not causally separate stated agreement and stated consequences for certain types of aid. Perceived consequences are likely causally prior to agreement to aid, but could as well, e.g., via motivated reasoning, be endogenous to support. We deem it still insightful to infer the overall structure of agreement and perceptions of more vs. less robust types of aid, as debated at the time of our survey, given these likely directly relate to domestic pressures to governments.

We want to thank R2 again for the insightful comments provided, which particularly contributed to fleshing out the theoretical contribution our manuscript makes.

Comments by Reviewer #3

R3-1 The study addresses a highly relevant issue by examining how public opinion in Western countries is formed and evolves with respect to support for Ukraine. This is a crucial contribution to understanding strategic decisions by Western allies in response to Russian aggression. The topic is highly relevant in the current geopolitical context, particularly for decision-making within NATO and the EU.

We thank R3 very much for his/her detailed review of our manuscript. We have carefully considered her/his comments and revised the manuscript according to his/her suggestions.

R3-2 A strong asset of this study is a fairly comprehensive survey conducted by the authors. The study employs a high-quality sampling strategy using YouGov's online panels in the United States, United Kingdom, Germany, France, and Italy. The sampling strategy and data collection methods appear to be robust and reflect the current distribution of political preferences in each country. The conjoint and vignette experimental designs are well structured and appropriate for addressing the research questions.

The results are presented clearly, with logical explanations of respondents' preferences and their variations. The use of conjoint experiments to quantitatively analyze the impact of different attributes on respondents' preferences is particularly noteworthy. The results effectively answer the research questions and demonstrate the significant role of public opinion in shaping Western support for Ukraine.

We would like to thank R3 for highlighting the strengths of our study.

R3-3 Er. The study conducts a good level of literature review, although there is not much literature available at the moment due to the ongoing situation. I think the authors adequately situate their findings within the existing body of research. By analyzing the relationship between public resolve and policy decisions from a new micro-foundational perspective, the study makes a certain level of theoretical contribution. This provides a good foundation for future research in this area.

We thank R3 for her/his remarks regarding the links to the existing literature. We would like to point out that we have elaborated on this point even further in the (new) SI section 1.

R3-4 The paper is generally well written and logically organized, with clear connections between sections. The flow of the research is smooth and the arguments are effectively communicated. However, some sections could benefit from additional elaboration for clarity.

We appreciate the comment from R3. We have revised important parts to further improve the understanding and highlight the relevant information, e.g., in the reworked framing of the introduction, the new argument section, or the new introduction to sections 3.1.3 and 3.2.

R3-5 The rationale for the selection of attribute levels and their operationalization could be more explicitly detailed. Providing more context on how these levels were determined would enhance the reader’s understanding. In addition, the study could be strengthened by comparing its findings with those of other relevant research, such as Eurobarometer surveys, which have conducted extensive surveys on the Russia-Ukraine conflict with more accurate and comprehensive demographic profiles. There is not much academic research on this topic because the war is still ongoing. However, there are tons of reports and news articles on public opinion and attitudes towards the Russian-Ukrainian war. The authors will be able to find additional insights from these views.

We thank R3 for her/his comment. Currently, we provide a detailed explanation of the selection of attribute levels in the methods section 2.1.1 (as well as in the pre-analysis plan), and refrained from repeating this extensively in the main manuscript body. In order to keep the main body light we did not expand these parts, but if deemed important by R3 we will of course do so.

We are grateful that R3 pointed us to previously published surveys. We have taken this up in the introduction (*Descriptive evidence from surveys with European citizens indicates that Western populations show nuanced preferences [36].*¹⁰) and in footnote 29, which reads as:

The fact that we find a statistically significant spread between the countries in the attributes on concessions and sovereignty (especially between the UK and Italy) confirms the findings of Thomson et al. [45] and Politico (<https://www.politico.eu/article/european-an-s-think-ukraine-lose-war-russia-survey>). Interestingly, the particularly pronounced resolve of the British public (also in comparison to the Italian public) can already be seen in surveys conducted shortly after the start of the war (see e.g. YouGov on February 28th, 2022: <https://yougov.co.uk/politics/articles/41276-european-reaction-russian-invasion-ukraine>). However, what these surveys cannot show (compared to our study) are the trade-offs that citizens make.

R3-6 The authors claim that this is the first large-scale cross-national study of public attitudes toward support for Ukraine. However, similar surveys have been conducted by think tanks such as the Chicago Council on Global Affairs and Pew Research. Acknowledging and differentiating from these existing studies would provide a clearer positioning of the current research.

We are grateful for R3 pointing out these studies. However, we also note that while it is insightful to compare to these studies, they are regularly single country studies (with a heavy US focus) or observational (with higher potential for bias, and lacking an assessment of actual trade-offs). We included this in the (new) SI section 1, which reads as:

¹⁰See also Ash *et al.* [37].

... of course, besides scholarly work, there also exists opinion polling by the Chicago Council on Global Affairs and by Pew Research Center. The former only investigate US public opinion, on which we relevantly expand in our five-country study. The latter regularly conducts cross-country opinion polls. The former (e.g., [46]) report strong majority support from US citizens' for Ukraine aid – a finding likely overstated due to social desirability bias, given earlier surveys indicated relevant sensitivity to costs and security considerations among US citizens [47], which can not be adequately captured in non-experimental surveys. Similarly, while cross-country opinion polling indicates split public opinions (e.g., [48]) on Ukraine aid, these studies cannot adequately provide evidence on the trade-offs involved to assess the bounds of citizens' actual resolve.

Taken together, none of these studies uses survey-experimental methods. Our conjoint and vignette experiments outperform, however, standard approaches as they very well approximate real-world decision-making of citizens [1], as they are effective at mitigating social desirability bias [5], a very relevant concern in public opinion surveys¹¹. Even more, our survey-experimental research design is able to simultaneously, and causally, investigate the trade-offs among the three core dimensions of Ukraine support as present in public debate – normative, economic, and security-related considerations – while, again with a causal inference design, assessing agreement with concrete military and economic aid types and perceptions related to these.

R3-7 The authors mentioned that the survey included respondents' characteristics (age, level of education, political orientation, etc.), but such details are completely absent from the analysis presented in the paper. It is well known that individual perceptions and views on the Russia-Ukraine war are significantly influenced by respondents' characteristics. The high inflation caused by the Russia-Ukraine war, which has led to a decline in the purchasing power of European citizens and a general economic downturn, is well known. In addition, the overall rise in prices, including energy and food costs, has significantly altered the political landscape in Europe, contributing to increased support for far-right parties. During economic recessions caused by supply shocks, low-income groups tend to suffer more. Therefore, perceptions of the Russia-Ukraine war and willingness to support Ukraine are strongly influenced by individuals' economic situation. This study does not address these issues in the main text.

We want to thank R3 for this legitimate comment. We agree that it is reasonable to hypothesize associations as indicated by R3. But while we could investigate this heterogeneity, in our opinion, conducting standard subgroup analysis for this large number of different subgroups would be too extensive given our already very dense manuscript. If going this route, one option would be to use recently proposed (machine learning) methods, for example

¹¹For example, observational studies on Russian public opinion after the war in Ukraine [49, 50] are likely severely overstating support for the regime [51]. Similar mechanisms are generally in play in all public opinion studies, however.

Goplerud *et al.* [52] or Robinson & Duch [53]. However, these are explorative approaches and do not allow for hypothesis testing.

Please also note that some of the respondent’s characteristics outlined above are used in the correlation analysis for Anti-Western attitudes (see SI Table 4). The results are also taken up in section 3.1.3, where we write:

we find that anti-western attitudes are relevantly associated [...] higher with worry over the personal or country-wide state of the economy [...]; there is no discernible association with gender or work status (ceteris paribus).

Last, we want to highlight that the additional heterogeneity proposed by R3 can be assessed with our replication data in future research.

R3-8 In addition, the survey should take into account whether respondents are natives or immigrants, which is also not mentioned in the study. Presenting survey results without distinguishing between economic and social groups can easily lead to biased results. Could these factors be the reason why support for Ukraine is not unconditional?

We agree with R3 that there can theoretically be a certain degree of effect heterogeneity when comparing natives with immigrants. However, our sample only includes citizens of a country eligible to vote. Hence, any immigrants not naturalized are by definition excluded. We selected the sample in purpose to zoom onto the population relevant for domestic political decision making via voting. Additionally, we did not ask about the migration background of respondents in our survey. Of course, one can be a citizen and have immigrated some time in the past at the same time. Past research indicates that immigrants have a largely similar preference structure compared to natives [54], albeit to varying degrees [55], while foreign policy preferences could also be an outlier, e.g., due to ethnic group attachments [56]. However, we are unfortunately neither aware of research that studies this explicitly, nor can we bring tailored own data to this question. Note that there is one potential proxy in this regard, which is religion – as protestant and catholic faith is more prevalent among natives compared to immigrants in several of our countries. As can be seen from Appendix Table 4, both Orthodox Christian and Muslim faith is relevantly linked to anti-western attitudes, which in turn are linked to disregard of Ukraine war outcomes. However, we refrain from interpreting this with respect to immigration, as the actual migration status of these respondents is unclear. Given the share of respondents indicating Muslim (2.7%) or Orthodox christian (1.2%) faith is very small, this also does not drive overall outcomes.

R3-9 Overall, this study provides an important analysis of public opinion in Western countries regarding support for Ukraine. The methodology is sound and the results are clearly presented. However, the study would benefit from a more detailed explanation of the

data, comparisons with other research, acknowledgement of existing studies in the field, and consideration of the economic and social groups of respondents. By addressing these issues, the study could provide more robust and comprehensive insights into the dynamics of public opinion and policy implications.

Overall, we again want to thank R3 for the thorough review of our manuscript and the insightful comments we received, allowing us to relevantly improve our manuscript.

References

1. Hainmueller, J., Hangartner, D. & Yamamoto, T. Validating vignette and conjoint survey experiments against real-world behavior. *Proceedings of the National Academy of Sciences* **112**, 2395–2400 (2015).
2. Bansak, K., Hainmueller, J., Hopkins, D. J. & Yamamoto, T. Beyond the Breaking Point? Survey Satisficing in Conjoint Experiments. *Political Science Research and Methods* **9**, 53–71. ISSN: 2049-8470, 2049-8489. (2024) (Jan. 2021).
3. Hainmueller, J., Hopkins, D. J. & Yamamoto, T. Causal Inference in Conjoint Analysis: Understanding Multidimensional Choices via Stated Preference Experiments. **22**, 1–30 (2014).
4. Rudolph, L., Freitag, M. & Thurner, P. W. Deontological and consequentialist preferences towards arms exports: A comparative conjoint experiment in France and Germany. *European Journal of Political Research* **63**, 705–728 (2024).
5. Horiuchi, Y., Markovich, Z. & Yamamoto, T. Does conjoint analysis mitigate social desirability bias? *Political Analysis* **30**, 535–549 (2022).
6. Kertzer, J. D., Powers, K. E., Rathbun, B. C. & Iyer, R. Moral Support: How Moral Values Shape Foreign Policy Attitudes. *The Journal of Politics* **76**, 825–840 (2014).
7. Kertzer, J. D. & Zeitzoff, T. A Bottom-Up Theory of Public Opinion about Foreign Policy. *American Journal of Political Science* **61**, 543–558 (2017).
8. Kertzer, J. D. & Tingley, D. Political Psychology in International Relations: Beyond the Paradigms. *Annual Review of Political Science* **21**, 319–339 (2018).
9. Burstein, P. The Impact of Public Opinion on Public Policy: A Review and an Agenda. *Political Research Quarterly* **56**, 29–40 (2003).
10. Page, B. I. & Shapiro, R. Y. Effects of Public Opinion on Policy. *American Political Science Review* **77**, 175–190 (1983).
11. Wlezien, C. The Public as Thermostat: Dynamics of Preferences for Spending. *American Journal of Political Science* **39**, 981–1000 (1995).
12. Hager, A. & Hilbig, H. Does Public Opinion Affect Political Speech? *American Journal of Political Science* **64**, 921–937 (2020).
13. Bakaki, Z., Böhmelt, T. & Ward, H. The Triangular Relationship between Public Concern for Environmental Issues, Policy Output, and Media Attention. *Environmental Politics* **29**, 1157–1177 (Nov. 2020).
14. Schaffer, L. M., Oehl, B. & Bernauer, T. Are policymakers responsive to public demand in climate politics? *Journal of Public Policy* **42**, 136–164 (2022).
15. Chu, J. A. & Recchia, S. Does public opinion affect the preferences of foreign policy leaders? Experimental evidence from the UK parliament. *The Journal of Politics* **84**, 1874–1877 (2022).

16. Tomz, M., Weeks, J. L. & Yarhi-Milo, K. Public opinion and decisions about military force in democracies. *International Organization* **74**, 119–143 (2020).
17. Vandeweerdt, C., Kerremans, B. & Cohn, A. Climate voting in the US Congress: the power of public concern. *Environmental Politics* **25**, 268–288 (2016).
18. Anderson, B., Böhmelt, T. & Ward, H. Public opinion and environmental policy output: a cross-national analysis of energy policies in Europe. *Environmental Research Letters* **12**, 114011 (2017).
19. Weaver, A. A. Does protest behavior mediate the effects of public opinion on national environmental policies? A simple question and a complex answer. *International Journal of Sociology* **38**, 108–125 (2008).
20. Hyde, S. D. Experiments in international relations: Lab, survey, and field. *Annual Review of Political Science* **18**, 403–424 (2015).
21. Mutz, D. C. *Population-based survey experiments* (Princeton University Press, 2011).
22. Stantcheva, S. How to Run Surveys: A Guide to Creating Your Own Identifying Variation and Revealing the Invisible. *Annual Review of Economics* **15**, 205–234. ISSN: 1941-1383, 1941-1391. (2024) (Sept. 2023).
23. Harrison, G. W. in *Handbook of Choice Modelling* 246–275 (Edward Elgar Publishing, 2024).
24. Hainmueller, J., Hangartner, D. & Yamamoto, T. Validating Vignette and Conjoint Survey Experiments against Real-World Behavior. en. *Proceedings of the National Academy of Sciences* **112**, 2395–2400 (2015).
25. Berinsky, A. Assuming the Cost of War: Events, Elites, and American Public Opinion for Military Conflict. *The Journal of Politics* **69**, 975–997 (2007).
26. Gelpi, C. & Reifler, J. The Polls—Review: Reply to Berinsky and Druckman: Success Still Matters. *Public Opinion Quarterly* **72**, 125–133 (2008).
27. Ganter, F. Identification of preferences in forced-choice conjoint experiments: Reassessing the quantity of interest. *Political Analysis* **31**, 98–112 (2023).
28. Abramson, S. F., Kocak, K., Magazinnik, A. & Strezhnev, A. Detecting Preference Cycles in Forced-Choice Conjoint Experiments. *SocArXiv*. <https://osf.io/preprints/socarxiv/xjre9> (March 2024).
29. Lemke, T. & Habegger, M. W. Foreign interference and social media networks: A relational approach to studying contemporary Russian disinformation. *Journal of Global Security Studies* **7**, ogac004 (2022).
30. Guisinger, A. & Saunders, E. N. Mapping the boundaries of elite cues: How elites shape mass opinion across international issues. *International Studies Quarterly* **61**, 425–441 (2017).
31. Eichenberg, R. C. in *Oxford research encyclopedia of politics* (2016).

32. Kertzer, J. *Resolve in international politics* (Princeton University Press, 2016).
33. Berinsky, A. J. In Time of War: Understanding American Public Opinion from World War II to Iraq (2009).
34. Dill, J., Howlett, M. & Müller-Crepon, C. At Any Cost: How Ukrainians Think about Self-Defense Against Russia. *American Journal of Political Science* **68**, 1460–1478 (2024).
35. Marsh, N. Responding to needs: military aid to Ukraine during the first year after the 2022 invasion. *Defense & Security Analysis* **39**, 329–352 (2023).
36. Krastev, I. & Leonard, M. Fragile unity: Why Europeans are coming together on Ukraine (and what might drive them apart). *ECFR Policy Briefs* (2023).
37. Ash, T. G., Krastev, I. & Leonard, M. *United West, divided from the rest: Global public opinion one year into Russia's war on Ukraine*. (European Council on Foreign Relations, 2022).
38. Flores-Macías, G. A. & Kreps, S. E. Borrowing Support for War: The Effect of War Finance on Public Attitudes toward Conflict. *Journal of Conflict Resolution* **61**, 997–1020 (2017).
39. Scheve, K. F. & Slaughter, M. J. Labor Market Competition and Individual Preferences Over Immigration Policy. en. *Review of Economics and Statistics* **83**, 133–145 (2001).
40. Heinrich, T. & Kobayashi, Y. How Do People Evaluate Foreign Aid To 'Nasty' Regimes? *British Journal of Political Science* **50**, 103–127 (2020).
41. Kreps, S. & Maxey, S. Mechanisms of Morality: Sources of Support for Humanitarian Intervention. *Journal of Conflict Resolution* **62**, 1814–1842. (2020) (2018).
42. Dill, J. & Schubiger, L. I. Attitudes toward the use of force: Instrumental imperatives, moral principles, and international law. *American Journal of Political Science* **65**, 612–633 (2021).
43. Dill, J., Sagan, S. D. & Valentino, B. A. Inconstant Care: Public Attitudes Towards Force Protection and Civilian Casualties in the United States, United Kingdom, and Israel. *Journal of Conflict Resolution* **67**, 587–616 (2023).
44. Börzel, T. A. & Zürn, M. Contestations of the liberal international order: From liberal multilateralism to postnational liberalism. *International organization* **75**, 282–305 (2021).
45. Thomson, C., Mader, M., Münchow, F., Reifler, J. & Schoen, H. European public opinion: united in supporting Ukraine, divided on the future of NATO. *International Affairs* **99**, 2485–2500 (2023).
46. Chicago Council on Global Affairs. *Americans See High Stakes for Western Security in Russia-Ukraine War* Accessed: 2024-10-04. 2024. <https://globalaffairs.org/research/public-opinion-survey/americans-see-high-stakes-western-security-russia-ukraine-war>.

47. Chicago Council on Global Affairs. *American Public Support for Assistance to Ukraine Has Waned but Remains* Accessed: 2024-10-04. 2023. <https://globalaffairs.org/research/public-opinion-survey/american-public-support-assistance-ukraine-has-waned-still>.
48. Pew Research Center. *NATO Seen Favorably in Member States, Confidence in Zelenskyy Down in Europe, U.S.* Accessed: 2024-10-04. 2024. https://www.pewresearch.org/global/2024/07/02/nato-seen-favorably-in-member-states-confidence-in-zelenskyy-down-in-europe-us/gap_2024-07-02_russia-nato_00_03/.
49. Ferraro, V. Why Russia invaded Ukraine and how wars benefit autocrats: The domestic sources of the Russo-Ukrainian War. *International Political Science Review* **45**, 170–191 (2024).
50. Kizilova, K. & Norris, P. “Rally around the flag” effects in the Russian–Ukrainian war. *European Political Science* **23**, 234–250 (2024).
51. Foa, R. S. & Nezi, R. Piercing the Fog of War: Measuring Russian Public Opinion via Online Search Data (2023).
52. Goplerud, M., Imai, K. & Pashley, N. E. *Estimating Heterogeneous Causal Effects of High-Dimensional Treatments: Application to Conjoint Analysis* June 2024. arXiv: [2201.01357 \[stat\]](https://arxiv.org/abs/2201.01357). (2024).
53. Robinson, T. S. & Duch, R. M. How to Detect Heterogeneity in Conjoint Experiments. *The Journal of Politics* **86**, 412–427. ISSN: 0022-3816. (2024) (Apr. 2024).
54. Dancygier, R. & Saunders, E. N. A new electorate? Comparing preferences and partisanship between immigrants and natives. *American Journal of Political Science* **50**, 962–981 (2006).
55. Politi, E., Chipeaux, M., Lorenzi-Cioldi, F. & Staerklé, C. More royalist than the king? Immigration policy attitudes among naturalized citizens. *Political Psychology* **41**, 607–625 (2020).
56. Smith, T. *Foreign attachments: The power of ethnic groups in the making of American foreign policy* (Harvard University Press, 2000).

Response Memo # 2

Citizen's Resolve Against Autocratic Aggression: Survey Experimental Evidence from Five NATO Countries on Supporting Ukraine

March 29, 2025

Dear Referees,

We thank you for the work entailed in reassessing the revised version of our manuscript NCOMMS-24-39785A-Z “Citizen’s Resolve Against Autocratic Aggression: Survey Experimental Evidence from Five NATO Countries on Supporting Ukraine”. We are grateful for your very helpful feedback on the previous version of our manuscript. In light of your suggestions, we have carefully updated our manuscript regarding recent changes in societal debates and new literature. Of course, we remain firmly committed to implementing whatever additional changes might be necessary to make the paper acceptable for publication in Nature Communications.

In what follows, we reproduce the round 2 comments as given by the reviewers to shortly respond to them directly (in **bold** font) and discuss the respective changes made in the manuscript. Also, we attach a version delineating track changes compared to the original submission. You find our replies to the comments of Reviewer #1 from page 2, of Reviewer #2 from page 4, and of Reviewer #3 from page 5.

Yours sincerely,

The authors

Comments by Reviewer #1

R1-1 I write with regard to your question regarding the appeal for the rejected manuscript, “Citizen’s Resolve Against Autocratic Aggression: Survey Experimental Evidence from Five NATO Countries on Supporting Ukraine.”

I read over the response memo while I do think that they were responsive to my concerns and suggestions, the revisions still leave my overall assessment of the paper unchanged.

I remain somewhat unenthusiastic about the suitability of the placement of this paper in a top general cross-field journal, like Nature Communications. It’s a fine empirical paper with original data, but in the scope of things, I think it is much more appropriate for a subfield-specific journal in political science.

Overall, we again want to cordially thank R1 for the feedback provided throughout the course of the revision process. Incorporating these comments has strengthened our manuscript considerably, as acknowledged by R1. We also note, however, that we do not agree with R1’s assessment of our manuscript after we conducted revisions.

In round 1, R1 concluded on our manuscript that “the advancement of knowledge through empirical work is a worth endeavor” [and...] “publication in Nature Communication is a decision that the editor must make.” In the judgment of R1, our manuscript hence makes a substantial empirical contribution: hard scientific evidence, to our knowledge the very first cross-country survey-experimental study, on mass preferences on support for a war fought on foreign ground – one of the few large-scale cross-country experimental investigations of foreign policy preferences with a theoretically novel and societally contested issue as application.¹

Moreover, after revisions, R2 and R3 agree on both the empirical and theoretical contribution our manuscript makes. For example, in the words of R2, we provide “an important contribution [...] to the study of resolve in IR and the study of mass attitudes on war in comp gov”.

To summarize, with respect to theory: We provide the very first extension of the war support literature to the case of support for a war fought on foreign soil. We provide a pre-registered argument for how this changes citizens’ cost-benefit calculus, especially regarding the role of domestic economic costs and human suffering – and is, therefore, a broader contribution to how citizens assess war costs beyond the domestic realm. Additionally, we contribute by comparing preference formation in the 5 major military powers of the West. Theoretically, it is unclear whether findings from the much-studied but in

¹Evidence currently published mainly stems from single-country public opinion polls, prone to social desirability biases as preferences on Ukraine strategy are, at the core, constituted from multi-dimensional valence issues.

many aspects specific US case generalize to other contexts – which we can confirm. Moreover, we extend the general foreign policy literature by theorizing a value foundation rooted in the rule-based liberal world order as a core but hitherto unnoted moderator of foreign policy preferences. Last, our study is replicating and extending an impactful study on preferences of the Ukrainian population on Ukraine strategy (Dill et al., 2024, in AJPS), allowing to contrast how the directly affected Ukrainian and the indirectly affected US/European populations relate to the very same conflict. Again, something that has not been studied before.

Of course, we want to preclude that the difference in assessments by R1 and R2/R3 lies in us not communicating our theoretical contribution clearly in our manuscript. Therefore, we again carefully rewrote our abstract, introduction and conclusion sections, to improve and flesh out the communication of our theoretical contributions. Please refer to the track-changed version of our manuscript, which now more clearly communicates the above-mentioned contributions.

Taken together, we believe we make an, in our eyes and in the eyes of R2 and R3, “important advance of significance to *specialists* within [our] field” [our emphasis; see Nature Communications aims & scope webpage], and hence in line with the aims of Nature Communications.

Moreover, we believe our contribution is not only of interest to *specialists* in the field, but to the *general* scientific community, policy makers, and public. We strongly emphasize this point as R1 seems not to have considered the societal relevance of our manuscript in her/his decision. With respect to societal debates, after Western democratic leaders promised aid ‘for as long as it takes’ early into the war, the issue is now societally and electorally strongly contested – as evident from current news and social media timelines. However, we know of no experimental studies that investigate citizen preferences on this crucial topic – even though it is the public that sets the mandates for Ukraine aid. Notably, our study allows us to assess to what extent the public in the five major Western NATO countries pressures their governments into different directions regarding Ukraine strategy and whether President Trump has a US majority mandate for the way he currently tries to resolve Ukraine-aid trade-offs. This is something no prior observational research can show and provides guidance for impactful societal debates regarding peace in Europe and unity among the Western countries we live in. To make this contribution more prominent, we updated and extended our case discussion (see also R2). Again, we see such a contribution to societal debates as fully in line with Nature Communications, publishing “research that is rigorous, reproducible and *impactful*” [our highlighting; see Nat. Comm. editorial values webpage].

Comments by Reviewer #2

R2-1 The authors have done a commendable job in making the framing of the paper more compelling, explaining the vignette experiment better and anchoring the paper better in the existing literature. I am still not sure that calling what they study "a micro-foundation of resolve" is completely accurate or necessarily the best way of explaining the theoretical angle of the paper, but I am convinced that the paper has an important contribution to make to the study of resolve in IR and the study of mass attitudes on war in comp gov. I also think the paper could hardly be more timely.

Of course, given that the appeal of the paper is bound up with its current political relevance, it will be critical for the authors to add a brief reflection on recent developments. In a sense the likely/imminent withdrawal of US aid from Ukraine seems to be a phenomenon largely disconnected from public opinion dynamics in the United States. And yet, the reasoning of the US administration has a decidedly populist flavor, ostensibly appealing to the need to "stand up for the American people".

So in short, I do not think this detracts from the value of the paper, which I would be enthusiastic to see in print, but it should briefly be addressed. Along similar lines, a replication of the AJPS study of Ukrainian attitudes that the authors rely on now suggests that their attitudes remain steadfast along some lines, but have changed/weakened along others. It might be worth gesturing to this as well.

We want to thank R2 again for the insightful comments provided, which particularly contributed to fleshing out the theoretical contribution our manuscript makes. We are thrilled R2 is "enthusiastic" to see our manuscript in print, and that R2 now sees more value in our argument, making "an important contribution to [...] the study of resolve in IR and the study of mass attitudes on war in comp gov."

We fully agree that the events that unfolded after January 8 (and after we submitted the last version) need to be taken into account when discussing our findings. Our manuscript adds important nuance to very relevant societal debates – basically emphasizing A) the structural consensus among the populations of top NATO countries, even though their governments currently find no common ground. And B) that US President Trump has no majority mandate for the way he solves the trade-offs on Ukraine strategy. We now reflect on these thoughts in the updated introduction, discussion, and case section (see track-changed version). In the conclusion, we emphasize particularly the role of domestic institutions for translating preferences into representation, setting differentially strict bounds on how strongly the quarter of the population we observe as indifferent regarding Ukraine outcomes can reflect in government strategy.

Also, we thank R2 for pointing us to the new working paper on preferences of the Ukrainian population (which we found on the webpage of one of this study's authors as a working paper). This is now also taken up in the introduction and discussion sections.

Comments by Reviewer #3

R3-1 I have reviewed the authors' response to my comments and their revised manuscript, and think they have made a sincere effort to address the key points while maintaining the overall focus and structure of their study. They have clarified their methodological choices, particularly in explaining the choice of attributes and their operationalization, and they have also acknowledged relevant prior research, including comparisons with existing studies. In addition, they have improved the clarity of their argument by refining sections such as the introduction and methodology, making the manuscript more coherent and structured.

While some areas, such as subgroup analyses based on economic or social background, could be further explored in future research, I find that the authors have provided reasonable justifications for their current approach. Their revisions adequately strengthen the manuscript without altering its core contributions. Given these improvements, I think the paper is now ready for publication.

Overall, we again want to thank R3 for the thorough review of our manuscript and the insightful comments we received, allowing us to relevantly improve our manuscript. We are thrilled that R3 assesses our manuscript as “ready for publication”, praising the “improved the clarity of [our] argument” and “justifications for [our] current approach”.